# A novel model–data fusion approach to terrestrial carbon cycle reanalysis across the contiguous U.S using SIPNET and PEcAn state data assimilation system v. 1.7.2

Hamze Dokoohaki[1], Bailey D. Morrison[2], Ann Raiho[3], Shawn P. Serbin[2], and Michael Dietze[4]

[1]University of Illinois at Urbana-Champaign, Crop Science Department, Urbana-Champaign, IL, USA
[2]Brookhaven National Laboratory, Environmental and Climate Sciences Department, Upton, NY, USA
[3]Colorado State University, Fort Collins, CO, USA
[4]Boston University, Earth and Environment Department, Boston, MA, USA

**Correspondence:** Hamze Dokoohaki (hamzed@illinois.edu)

**Abstract.** The ability to monitor, understand, and predict the dynamics of the terrestrial carbon cycle requires the capacity to robustly and coherently synthesize multiple streams of information that each provide partial information about different pools and fluxes. In this study, we introduce a new terrestrial carbon cycle data assimilation system, built on the PEcAn model-data eco-informatics system, and its application for the development of a proof-of-concept carbon "reanalysis" product that harmonizes carbon pools (leaf, wood, soil) and fluxes (GPP, Ra, Rh, NEE) across the contiguous United States from 1986-2019. We first calibrated this system against plant trait and flux tower Net Ecosystem Exchange (NEE) using a novel emulated hierarchical Bayesian approach. Next, we extended the Tobit-Wishart Ensemble Filter (TWEnF) State Data Assimilation (SDA) framework, a generalization of the common Ensemble Kalman Filter which accounts for censored data and provides a fully Bayesian estimate of model process error, to a regional-scale system with a calibrated localization. Combined with additional workflows for propagating parameter, initial condition, and driver uncertainty, this represents the most complete and robust uncertainty accounting available for terrestrial carbon models. Our initial reanalysis was run on an irregular grid of $\sim 500$ points selected using a stratified sampling method to efficiently capture environmental heterogeneity. Remotely sensed observations of aboveground biomass (Landsat LandTrendr) and LAI (MODIS MOD15) were sequentially assimilated into the SIPNET model. Reanalysis soil carbon, which was indirectly constrained based on modeled covariances, showed general agreement with SoilGrids, an independent soil carbon data product. Reanalysis NEE, which was constrained based on posterior ensemble weights, also showed good agreement with eddy flux tower NEE and reduced RMSE compared to the calibrated forecast. Ultimately, PEcAn's carbon cycle reanalysis provides a scalable framework for harmonizing multiple data constraints and providing a uniform synthetic platform for carbon monitoring, reporting, and verification (MRV) and accelerating terrestrial carbon cycle research.

## 1 Introduction

Accurate assessment of both biogenic carbon stocks and exchanges between the biosphere and atmosphere is crucial for a more complete carbon monitoring, reporting, and verification (MRV) framework, as well as understanding climate-carbon feedbacks.

However quantifying different components of the carbon cycle to identify whether different regions/landscapes are C sinks or sources has been a major challenge for earth scientists (Williams et al., 2005) as the sink strength of the terrestrial biosphere is more variable than the ocean (Battle et al., 2000). For this reason, investigating the current and future state of the terrestrial carbon cycle has gained considerable scientific attention.

Substantial time and energy has been devoted to developing terrestrial carbon data products aimed at MRV and improving our understanding of the carbon cycle at broader spatial scales. Some of these products are based on up-scaled field measurements that interpolate between observations, such forest biomass maps based on inventory plots (Pan et al., 2011), soil carbon maps based on soil cores (Mikhailova et al., 2016), and carbon flux maps based on eddy covariance (Xiao et al., 2014). Other products are based on calibrating remote sensing measurements against in-situ data, such as leaf area index (Liu et al., 2018) and aboveground biomass (Myneni et al., 2001), or quantify properties that correlate strongly with carbon stocks or changes in those stocks, such as land use/land cover (Schillaci et al., 2017) and disturbance (Liu et al., 2011). These data products describe particular processes or features of the terrestrial carbon cycle at various spatial scales and have been essential to improving our understating of plant and soil processes. However, each of these products reflect a single pool or flux, and thus is only showing us part of the picture about the terrestrial carbon cycle. It is a highly challenging task to combine this wealth of observational information into a coherent and comprehensible view on carbon cycle and assure consistency between the available data products (Cavallaro et al., 2018). Furthermore, many parts of the carbon cycle are hard to observe at large scales, and our understanding of the processes that connect different pools and fluxes is often limited to smaller numbers of intensively studied sites rather than large-scale monitoring. Therefore, often times, the full potential of these data sets, as well as their combination, is not fully exploited (Montzka et al., 2012).

An alternative to the piece-wise understanding provided by derived data products is to use process-based terrestrial biosphere models (TBM) to quantify and understand carbon cycle dynamics holistically. The strength of TBMs lie in their ability to represent the carbon cycle in an internally-consistent manner that conserves mass and captures our current understanding of physical, chemical, and biological processes. However, their weakness comes from the many uncertainties that are present in such models, most of which are not formally accounted for in current applications (Raiho et al., 2020; Dietze, 2017). Most existing terrestrial models have been developed using deterministic strategies in which uncertainties associated with model drivers, model parameters, and initial conditions are ignored (Peylin et al., 2016). In addition, these models can have considerable amounts of process error, which arises from a combination of inherent stochasticity (e.g. disturbance, mortality, dispersal), heterogeneity in ecosystem processes, and structural uncertainties that reflect our imperfect knowledge about how to best represent system processes (Williams et al., 2005). The result is models that are not well-constrained and are highly prone to producing large errors (Scholze et al., 2017). Because of this, it is critical that we properly propagate uncertainties into TBM predictions and continuously confront those predictions with observational data to reduce the uncertainties around TBMs predictions (Scholze et al., 2017).

Rather than relying on models or data alone, data assimilation techniques offer a method for combining their strengths and addressing their weaknesses. Data Assimilation (DA) refers to a robust mathematical framework for constraining model predictions with observational data where the intellectual motivation is to construct an integrated view of a system. Data

assimilation is fundamentally a data-driven exercise focused on the synthesis of information across different data streams, and provides a statistical framework for linking data sources with different temporal and spatial resolutions to generate a composite data product that contains significantly more information than the individual data sources (Dietze, 2017). Unlike statistical data fusion approaches, data assimilation can also link together data about different pools and fluxes by leveraging the process understanding that is embedded in models, which in turn is often linked to intensively studied sites. In a data assimilation system, we can even exploit non-carbon observations to constrain the carbon cycle indirectly through the relations implemented in the process model (Scholze et al., 2017). At the same time, data assimilation systems act to continually constrain model uncertainties and continually pull models back to what actually occurred, rather than the many possible futures models may envision. As such, DA is distinctly different from, and potentially more powerful than, either forward simulations of calibrated models or derived data products that transform observations through statistical models alone.

When run as a hindcast, a frequent goal of data assimilation is to produce a 'reanalysis' product. The goal of a reanalysis product is to provide a synthetic, harmonized best-estimate of past pools and fluxes across space and time. Reanalysis products are popular in many parts of the Earth system sciences, such as harmonized climate reanalysis products generated by rerunning weather models constrained by historical observations. Reanalysis products are studied directly to better understand system properties, processes, and spatiotemporal variability and serve as inputs to other models and analyses. However, despite their clear potential to improve MRV and scientific understanding, there has been little effort to develop large-scale reanalysis products for the terrestrial carbon cycle. Our goal in this paper is to demonstrate an initial 'proof-of-concept' for a national-scale terrestrial carbon reanalysis that spans the contiguous US over a multi-decadal period (1986-2019) where remotely-sensed data constraints are available at a large scale.

Among data assimilation approaches to generate a carbon cycle reanalysis, Sequential Data Assimilation (SDA), in which new observations are assimilated iteratively through time, holds particular promise for combining a suite of datasets available on the terrestrial carbon cycle. There are a number of studies, predominately at the site scale, examining different approaches and applications of DA systems in carbon cycle assessment. For example, (Viskari et al., 2020) used SDA to continuously update the Yasso15 model's state variables to improve soil organic carbon estimates. Similarly, (Viskari et al., 2015) used tower and satellite observations of phenological state to improve seasonal carbon fluxes in the ED2 model. Fox et al. (2018) assimilated LAI and biomass into the CLM4.5 model using ensemble adjustment Kalman filter and showed that model carbon forecasts in central New Mexico were improved on monthly and longer timescales. (sch) introduced the Max Planck Institute Carbon Cycle Data Assimilation System (MPI-CCDAS) built around JSBACH model. They used globally distributed FAPAR observations and atmospheric CO2 to simulate phenology and net land carbon balance. In another study, (Chen et al., 2008) developed a joint parameter and state data assimilation method called SEnKF that not only dramatically reduced the uncertainty in state variables, but also accounted for the temporal evolution of model parameters. In a site-level study Gao et al. (2011) used Ensemble Kalman Filter (EnKF) in a data assimilation scheme to forecast the carbon cycle. Overall, these studies confirm that EnKF can effectively assimilate multiple data sets into an TBM.

While initial applications of SDA to the terrestrial carbon cycle have been promising, they have had a number of important limitations. Like other modeling studies, they have relied on an incomplete accounting of model uncertainty. For example, some

data assimilation studies have relied on meteorological driver uncertainty to generate ensemble spread (Fox et al., 2018) or spin-ups to represent initial condition uncertainty, and frequently, these studies ignore parameter uncertainty or model process error. Here, we build on a decade of work on uncertainty propagation and analysis within the PEcAn model-data informatics system (LeBauer et al., 2013; Fer et al., 2020) to generate the most complete and robust uncertainty accounting available to date (Raiho et al., 2020). At a high-level, we use an ensemble-based approach to propagate data-driven uncertainty in model drivers, initial conditions, and parameters, with parameter distributions derived from an across-site hierarchical Bayesian calibration against Ameriflux NEE and plant trait data. In addition, within the SDA we rely on the new Tobit-Wishart Ensemble Filter (TWEnF) to dynamically update and propagate a fully Bayesian estimate of the entire process error covariance matrix. The TWEnF also relaxes the strong Gaussian assumption behind most data assimilation methods, allowing for both zero-truncation (no negative values) and zero-inflation (more zeros than expected by the Gaussian), both of which are common features of ecological data.

In this study, we focus on scaling-up our previous site-level SDA (Raiho et al., 2020) to a national-scale terrestrial carbon re-analysis. In doing so we developed, tested, and calibrated a spatial localization algorithm for our covariance matrix (Petrie and Dance, 2010), estimating a 500km distance threshold beyond which covariances are set to zero to avoid spurious correlations. This system can run on a irregular spatial grid that uses cluster analyses to optimally distribute points in a multivariate environmental space (i.e., temperature, precipitation, elevation, AET, climatic water deficit) that reduces computational costs and better captures complex terrain than a regular grid. We also extended the TWEnF to calculate and save the spatiotemporally-varying posterior weights of each ensemble member, which allows us to infer sub-daily fluxes (e.g. GPP, NEE) across the simulation region even though our assimilation was run at an annual timescale. Having developed this system, we then report on our initial proof-of-concept reanalysis, which was constrained by MODIS leaf area index (LAI) and LandTrendr Aboveground Biomass (AGB). Finally, we perform an initial validation against Ameriflux NEE and SoilGrids soil organic carbon.

## 2 Material and Methods

### 2.1 PEcAn Platform and SIPNET model

We developed our regional state data assimilation framework within an ecological model-data informatics system, the Predictive Ecosystem Analyzer (PEcAn) v. 1.7.2 and under a module named assim.sequential. PEcAn is a toolbox that provides a unified format and workflow for processing inputs and outputs, running models, and performing analyses for a wide range of ecological models. PEcAn tools include sensitivity and uncertainty analysis, benchmarking, and parameter and state data assimilation (LeBauer et al., 2013; Fer et al., 2018; Raiho et al., 2020; Fer et al., 2020). PEcAn also provides a robust procedure for sampling and propagating uncertainties by generating ensembles across initial condition, model parameters and model drivers (soil and meteorological forcing). All methodological developments and tool improvements introduced in this paper have been added to PEcAn, which is available at(https://doi.org/10.5281/zenodo.5557914) and on Github (https://github.com/pecanproject/pecan/), as a virtual machine (https://opensource.ncsa.illinois.edu/projects/artifacts.php?key=PECAN), or as a series of Docker containers (https://hub.docker.com/u/pecan). PEcAn documentation is available through the project webpage (https://pecanproject.org). Specifically pull requests #2045, #2481, #2293, #2233, and #2066 on the pecan Github repository (https://github.com/pecanproject/pecan/) contains the largest contribution of this work to the pecan project.

We used the Simplified Photosynthesis and Evapotranspiration model (SIPNET) model (Braswell et al., 2005) within the PEcAn platform v. 1.7.2 as the process model for our regional data assimilation exercise. SIPNET was chosen for this initial proof-of-concept as it is computationally efficient but provides a non-trivial representation of carbon pools and fluxes. SIPNET represents the carbon cycle as a series of pools and fluxes, where it accounts for two vegetation carbon pools (leaf and wood) and a single aggregated soil carbon pool (Fig 1). To reduce the number of parameters in the SIPNET model, it makes the assumption that carbon stored in the leaves remains constant throughout the growing season and therefore, all fluxes in the carbon cycle (Fig 1) only affects the plant wood carbon pool. In addition, rather than a complex carbon allocation and phenology model, SIPNET uses a simple time-based function to model the phenology where on a specific day of the year; all leaves appear or fall off in a single time step. The simple representation of carbon cycle in SIPNET provides the opportunity to fully constrain both model parameters and state variables.

### 2.2 Model Calibration

To efficiently capture variability in vegetation properties across the contiguous US (CONUS), we aggregated National Land Cover Database (NLCD) land cover (Homer et al., 2012) into four plant functional types (PFTs): deciduous forest, evergreen forest, mesic grassland and arid grassland/shrubland. For deciduous forest we rely on the previous calibration reported by (Fer et al., 2021). The same methods were used to calibrate the remaining PFTs against a combination of plant trait data and eddy-covariance, as described below.

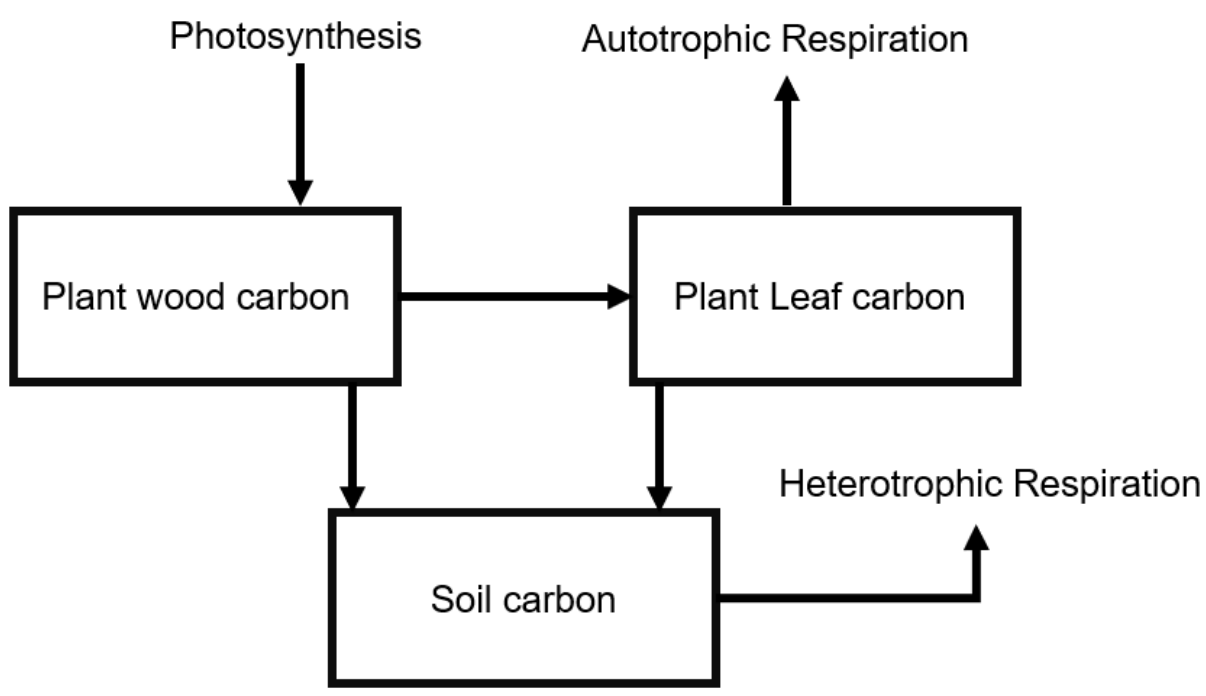

**Figure 1.** Carbon cycle representation by the SIPNET model

### 2.3 Priors and Trait Meta-analysis

PEcAn takes a fully-Bayesian approach to model calibration. Calibration begins with defining prior distributions for 35 parameters for the grass PFT and 32 parameters for the evergreen PFT. These priors were then updated using plant trait data in the BETY-db (LeBauer et al., 2018) by performing a hierarchical Bayesian meta-analysis following the methods described in (LeBauer et al., 2013). Trait data for 7 parameters (SLA, Amax, leafC, leaf respiration rate, leaf turnover rate, root respiration rate, root turnover rate) were available from 6 to 213 studies for different parameters. Meta-analysis posteriors were then used as informative priors in the subsequent sensitivity analysis and calibration, where they help ensure the selection of biologically plausible parameter estimates.

#### 2.3.1 Global Sensitivity Analysis (GSA)

Prior to model calibration, we performed global sensitivity analyses on the SIPNET model parameters at 5 grassland and 5 evergreen Ameriflux sites located across different ecoregions. The aim of this analysis was to reduce the dimensionality of the calibration by identifying the dominant parameters in the model. The informative prior distributions were sampled to generate 500 ensemble members to test the sensitivity of Net Ecosystem Exchange (NEE) to change in each parameter. Runs at each site were done for 15 years using gap-filled eddy flux tower meteorology as model drivers. A linear model between the simulated mean NEE and the sampled parameter values was used to decompose the variability around NEE and estimate the contribution of each parameter. The linear model included the main effect for each parameter as well as its first interactions. Main effects and total sensitivity indices was calculated using the Sums of Squares (SSQ) as follows (Dokoohaki et al., 2018; Wallach et al., 2019):

$$\text{Main effect sensitivity indices}: S_1 = \frac{SS_1}{SS_T}; S_2 = \frac{SS_2}{SS_T} \tag{1}$$

$$\text{Interaction sensitivity indices}: S_1 = \frac{SS_{12}}{SS_T} \tag{2}$$

$$\text{Total sensitivity indices}: S_1 = \frac{SS_1+SS_{12}}{SS_T}; S_2 = \frac{SS_2+SS_{12}}{SS_T} \tag{3}$$

Where $SS_1$ is the SSQ for the first parameter,$SS_{12}$is the SSQ of the interaction between first and second parameter, and $SS_T$ is the total SSQ. Based on the results of these analyses, parameters with a contribution of at least 5% in the total variability of NEE, were selected for further investigation and calibration for both conifer and Grass PFTs.

#### 2.3.2 Hierarchical parameter data assimilation

The hierarchical Bayesian parameter data assimilation (HPDA) framework within the PEcAn platform v. 1.7.2 was employed to constrain the most sensitive parameters of the SIPNET model found during the sensitivity analysis. We used HPDA to calibrate model parameters for a combined mesic and semiarid grassland PFT and an evergreen forest PFT against NEE observation from 6 eddy-covariance towers (Table 1). In contrast to a site-level model or single parameter single site calibration, hierarchical calibration accounts for site-to-site variability while also borrowing strength across sites during the parameter estimation procedure. Consequently, parameters are calibrated for all sites at the same time, where generally the estimated variability across sites identifies site sensitive parameters or missing processes in models (Fer et al., 2018).

Our calibration procedure closely followed Fer et al. (2018) by developing emulators of the likelihood surface at the site level. To develop the emulators, we used an adaptive sampling approach (Fer et al., 2018) to generate a $n$ dimensional grid over the parameter's space (where $n$ is the number of parameters) and then ran the SIPNET model at the n-dimensional points (e.g., knots) in this parameter set. Model runs were driven by gap-filled tower meteorology and varied in length from site-to-site depending on available data (Table 1). For each run, we then compared the model's predicted NEE to the observed NEE (30-min timestep, no gap-filling) and calculated a Likelihood score for each knot. Observations were filtered based on $u^*$ and effective sample size was corrected for autocorrelation following (Fer et al., 2018). Keeping with previous works (Fer et al., 2018; Reich and Cotter, 2015), we used an asymmetric heteroskedastic Laplace likelihood that accounts for the increase in observation error with the magnitude of the flux and error bias. In our case, these errors are larger in the positive direction (night-time respiration) than the negative. Site-level emulators were then developed by fitting a Gaussian Process (GP) model to the knots to construct a smooth n-dimensional likelihood surface in parameter space. After developing the site-level emulators, the hierarchical calibration was performed by MCMC. In the hierarchical MCMC, site-level parameters are accepted or rejected using the across-site mean and covariances as priors and using the emulator to predict the likelihood, while across-site mean and covariances are updated through Gibbs sampling (Fer et al., 2018):

$$\mu_{s,i} \sim MVN(\mu_g, \tau_g)$$
$$\mu_g \sim MVN(\mu_m, \tau_m)$$
$$\tau_g \sim W(n_g, V_g) \tag{4}$$

Where $\mu_g$ and $\tau_g$ represent the global mean parameter vector and the site-to-site covariance matrix while, $\mu_{s,i}$ are the $i$ site-level mean parameter vectors for each of $n$ sites. $\mu_g$ and $\tau_g$ were assigned uninformative multivariate Normal and Wishart priors, respectively.

Each calibration was run in three iterative rounds with adaptive sampling following (Fer et al., 2018), where the total number of proposed knots per round was set to $p \times 20$ ($p$ represents the number of parameters). In each round we used posterior mean to propose additional knots in the parts of parameter space needing additional detail, essentially behaving like a nested grid around the posterior mean. Finally, we used the Nash Sutcliffe model Efficiency (NSE) index and Root Mean Square Error (RMSE) to evaluate the performance of the model and examine whether the calibration has improved the model's capacity for simulating NEE following HPDA. NSE <= 1 indicates the goodness of fit of the simulated series against observations (1 = perfect fit).

### 2.4 Regional Site Selection

As previously noted, our data assimilation system runs on an irregular grid optimized to efficiently capture environmental variability. For the current 'proof-of-concept' analysis, we identified a total of 517 sites across CONUS by first selecting 46 Ameriflux sites that would be used to validate the assimilated fluxes, followed by 471 additional potential sites based on a cluster analysis of landcover class (NCLD) and climate variables. The cluster analysis used PRISM 800m precipitation, maximum and minimum temperature, and elevation, in addition to estimated actual evapotranspiration (AET), climatic water deficit (WD), total surface radiation, rain, and snow. Actual evapotranspiration and water deficit are biologically meaningful parameters that are well correlated with the distribution of vegetation types across multiple spatial scales (Stephenson, 1998). We began by using PRISM 800m elevation and produced monthly global total irradiance using the GRASS GIS (v7.8) for CONUS (Šúri and Hofierka, 2004). It was assumed total surface radiation remained consistent throughout the study time period as physical topography and solar distance change little at the decade time scale. To estimate AET, WD, rain, and snow, we collected 1 deg by 1 deg historical monthly wind vector data from CCSM4 (Danabasoglu et al., 2020) and estimated 1981-2010 monthly normals using methods reported in Morrison et al. (2019). Using the radiation and wind speed, along with maximum and minimum temperature, and precipitation, we derived climatic (not biological) AET and WD, which resulted in monthly climate normals for AET, WD, rain, and snow following methods described in Morrison et al. (2019). Next, a k-means sampling approach was used to determine the number of unique bioenvironmental clusters within each USGS Level 1 Ecoregion of CONUS (Omernik, 1987). Maximum and minimum temperature, radiation, AET, WD, rain, snow, and landcover class pixels for a given ecoregion were applied to a k-means algorithm. We performed multiple cluster analyses by increasing the value of k by one until at least two clusters were no longer distinct or overlapping one another. The number of unique clusters for an ecoregion was then determined by the maximum k value of distinct/non-overlapping cluster runs.

Lastly, each ecoregion's k value was used as a weight to determine the number of sites per ecoregion, generating the additional 471 randomly sampled sites to run in the SDA workflow, resulting in a total of 517 potential sites. Only sites with acceptable MODIS LAI QC values were used in the SDA (000-best result possible, 001-Good, very usable (Myneni and Park, 2015)), resulting in a total of 493 sites total to be included in the SDA simulations.

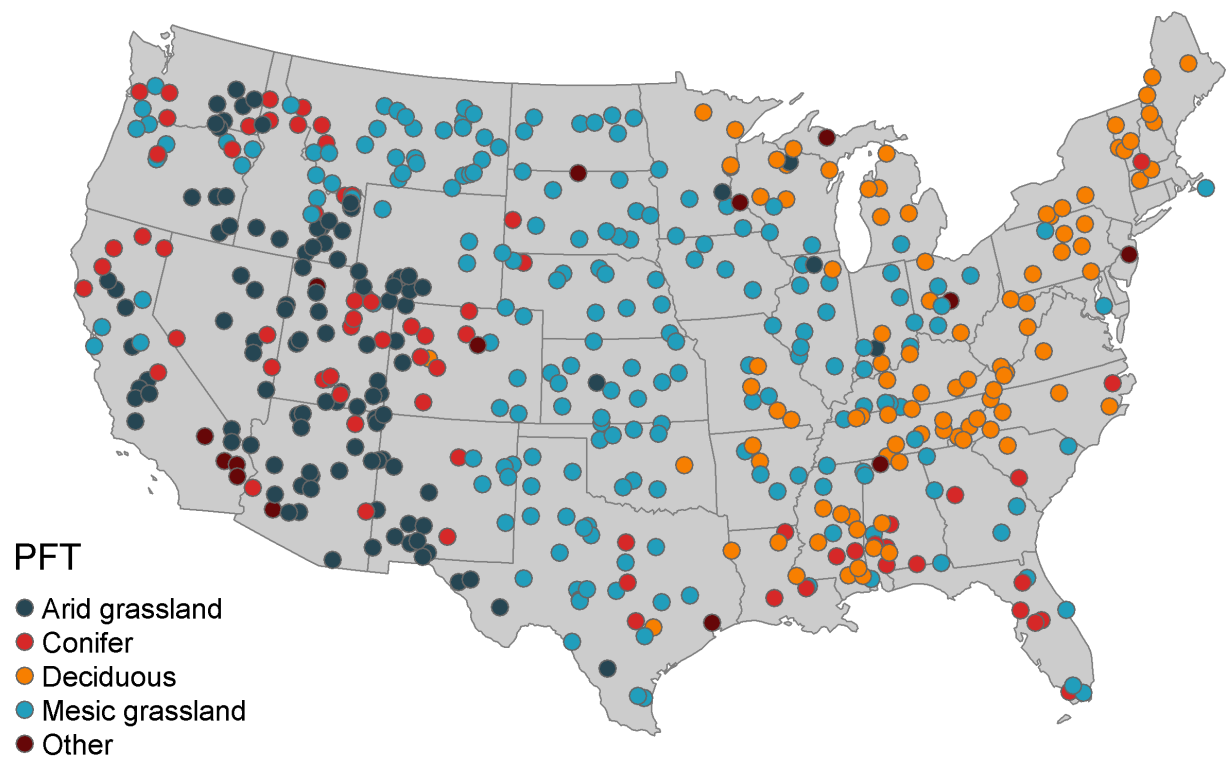

Figure 2. Distribution of sites with their corresponding plant functional type across CONUS

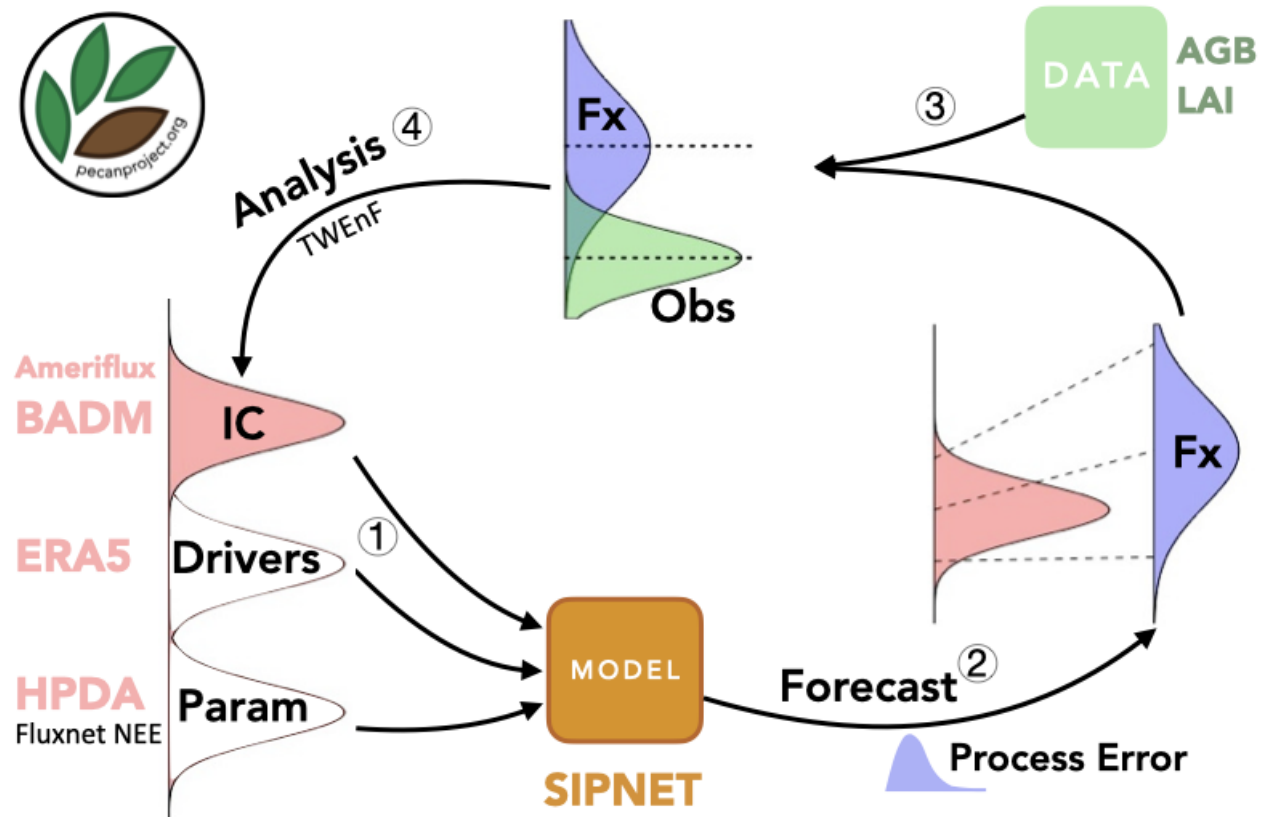

**Figure 3.** PEcAn State Data Assimilation system: Forecast/Analysis Cycle. Step1: Uncertainty propagation. Step2)Forward Forecast, Step3) Data preparation 4) Data assimilation

### 2.5 Sequential state Data Assimilation

The primary premise of sequential State Data Assimilation (SDA) methods is that neither models nor observations are perfect, but we can reduce uncertainties by finding a common ground between model simulations and multiple observations. SDA relies on an iterative cycle between forecasts and updates/analyses that leverage Bayes' theorem to update the forecast (prior) based on new observations (Likelihood) (Fig 3). For example, for a given site and time, prior to collecting data, model forecasts are the best estimates of the state of a system. The model's forecast can be summarized in terms of a mean vector of state variables, $\mu_f$, and an error covariance matrix, $P_f$. Once observations are made, we can update this prior using the newly observed data $Y$ (with associated observation error $R$) as a component of Bayes' Theorem to calculate a corrected estimate of state of a system $\mu_a$ and $P_a$. This update then serves as the initial conditions for further projections in time (Fig 4) (Dietze, 2017).

Classical data assimilation algorithms such Ensemble Kalman Filter (Evensen, 2009) have been used extensively in weather and ecological forecasting (Raiho et al., 2020; Viskari et al., 2020). However, most of these methods make strong assumptions about the probability distribution of the forecasts and observations and lack the ability to estimate the model process error. For example, the Gaussian assumption in the EnKF algorithm could generate negative soil carbon estimates and the omission of process error may lead to over confidence in model forecasts.

To account for censored state variables in our process model, we used the Tobit-Wishart Ensemble Filter (TWEnF) as described in (Raiho et al., 2020) to perform the analysis step within our data assimilation workflow (Fig 4). The forecast mean, covariance, and the likelihood function were transformed to the Tobit space to account for the left or right censored state variables (e.g., carbon pools cannot be negative). In addition, the prior for our process covariance matrix was assumed to be Wishart and prior shape parameters were stored and updated at each time step. This gave us the flexibility to not only relax the

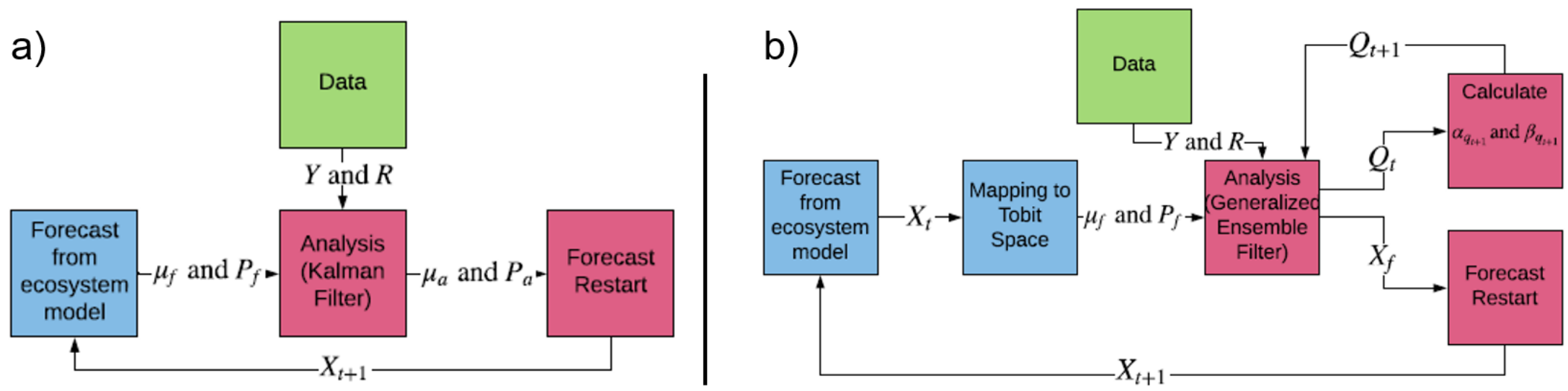

**Figure 4.** a) Classic Ensemble Kalman Filter method b) Tobit-Wishart Ensemble Filter; $X_f$: matrix of model forecasts; $\mu_f$ and $P_f$: Forecast mean and error covariance matrix ; $\mu_a$ and $P_a$: Analysis mean and error covariance matrix; $Y$ and $R$: Observation mean and error covariance matrix; $Q_t$ Model process error. Adopted from (Raiho et al., 2020)

Gaussian assumption behind the Kalman filter family of data assimilation methods but also to estimate and propagate model process error over the assimilation period.

The TWEnF algorithm presented in (Raiho et al., 2020) was expanded in this study to allow for regional data assimilation and multiple types of model process error $Q$ in this study as follows:

$$Y \sim \begin{cases} MVN(X,R) & > y_L \\ y_L & \leq y_L \end{cases} \tag{5}$$

$$X \sim MVN(x_f, Q) \tag{6}$$

$$x_f \sim MVN(\mu_f, P_f) \tag{7}$$

$$Q \sim Inv - Wishart(\alpha_q, \beta_q) \tag{8}$$

where $x_f$ and $X$ are the model estimates for each ensemble before and after accounting for process error, $Q$, respectively. $\alpha_q$ and $\beta_q$ are the parameters controlling the shape of the model process error $Q$ and $y_L$ represent the lower boundary for the censored observations, which is set to 0 in the current application.

By including the model error covariance estimates from all the sites in one unified model error covariance matrix in the update step, we can take the advantage of covariability between state variables in different sites. As a result, state variables with/without corresponding observations can be updated through their covariance with other variables in other sites. For example, this enables updates to LAI in a site with no direct observation, solely based on the covariance estimated with a nearby site with LAI observations. However, the extent to which we allow the sites to update their states could depend on some measure of dissimilarity such as physical or environmental distance between sites (Dietze, 2017). This problem in weather forecasting

literature is called the localization problem (Petrie and Dance, 2010). In this study, we set up the localization threshold to 500Km where the covariance between sites were adjusted using an exponential decay function as follows:

$$P_f = \exp\left(\frac{-1 \times d^2}{2 \times \nu^2}\right) \times P_f \quad (9)$$

where $d$ is the distance matrix between all the sites and $\nu$ is the cutoff distance where covariances will be push towards zero. More details on the localization setup can be found in Appendix A.

The new regional-scale generalization of the TWEnF analysis step was implemented in PEcAn v. 1.7.2 and fit in R statistical software (R Core Team, 2013) using the NIMBLE package (de Valpine et al., 2017). At every timestep, we ran three chains of MCMC, a total of 100,000 steps, and discarded the first 10,000 to account for burn-in. The primary variables of interest were the posterior estimates of $\mu_a$ and $P_a$ and the parameter controlling the covariance and shape of the process error ($Q$) while assuming the $R$ is known. We reduced the total number of unknown parameters by assuming a shared process error $Q$ among all sites.

Aboveground biomass (AGB) and Leaf Area Index (LAI) estimates were extracted from satellite imagery to act as our observational data to assimilate in the SDA algorithm. Estimates of the annual AGB and the pixel-wise standard deviation (SD) for each study site were collected from Landtrendr (Landsat-based detection of Trends in Disturbance and Recovery) at 30m resolution from 1983 to 2018 (Kennedy et al., 2018, 2010). LAI, SD, and QC values were collected from MODIS MOD15A2H (Moderate Resolution Imaging Spectroradiometer) at 500m resolution every 8 days from 2000-2019 for all sample sites (ORNL DAAC, 2018). LAI and SD observations with QC values that were not "Best result" (000) or "Good, very usable" (001) were removed from the LAI observation dataset and not used for assimilation (Myneni and Park, 2015). Since the assimilation was run on an annual basis, with the analysis occurring on July 15th, peak LAI was used as our annual LAI metric. Peak LAI was defined as the maximum LAI value within the 95th percentile for a single site and year. Similarly, the SD of peak LAI was defined as the corresponding SD value associated with the peak LAI value for each site/year. Any LAI SD value < 0.6 was reassigned a value of 0.6 following Viskari et al. (2015).

At the end of each assimilation cycle, we adjusted the updated state variables by using the ensemble adjustment technique (Anderson et al., 2009), rather than sampling new ensemble members from the analysis posterior, in order to maintain the covariance structure between states, drivers, and parameters. The goal of ensemble adjustment is to first shift the ensemble means to have the same mean as the posterior ($\mu_a$) and also linearly contract ensembles so they would also show the same variance as the posterior ($P_a$) (Anderson et al., 2009).

Our reanalysis was run with 20 ensemble members where each is associated with a different set of species parameters, meteorological drivers, and initial conditions. Parameter vectors for each ensemble member were drawn from the HPDA distribution for the across-site mean, $\mu_g$. The meteorological drivers covered the period from 1986-2019 and were resampled from the 10-ensemble member ERA5 reanalysis product, which has a 3-hr time step and a resolution of 0.5625 degrees (62km). Initial condition distributions were generated by resampling leaf, stem, and soil carbon pool data extracted from the Ameriflux Biological, Ancillary, Disturbance, and Management (BADM) database on an EPA L1 ecoregion basis.

### 2.6 Post hoc ensemble weight estimation

To focus our SDA on the model's state variables (carbon pools) and minimize the dimensions of $X$ for more efficient numerical sampling in Eq 5, we only included the main carbon cycle state variables (Aboveground biomass, leaf area index and soil carbon pool) in the $X$ matrix. Therefore, the model output variables that were not included in the $X$, such as carbon and water fluxes, were not adjusted during the Analysis step itself. Instead, these additional outputs were updated post-hoc for each site and time step by estimating the posterior probability of each ensemble member using the following equation:

$$P(X|\mu_a, P_a) \tag{10}$$

Equation 10 estimates the likelihood of producing the model simulations given the posterior (analysis) state of the system. Eq 10 provides a relative weight for each ensemble member that varies by year and site. A weighted mean and variance was then estimated for NEE, GPP, autotrophic respiration, and heterotrophic respiration using the estimated weights. These weights are applied not just to the cumulative sum of the fluxes (annual totals) but to the high-frequency time series between analysis time points. This approach was validated by comparing the reanalysis NEE to the Ameriflux observed NEE at a subset of points, with a more detailed validation occurring in a follow-up paper (Morrison et al. in prep).

### 2.7 Soil Carbon Validation

To assess our ability to indirectly infer soil carbon from assimilation of LAI and AGB into the SIPNET model, we compared our SOC estimates at all sites against the the soilGrids dataset (Hengl et al., 2017). SoilGrids is an interpolated data product at 250 m resolution produced using machine learning models by taking advantage of global soil profile information and a series of covariate data. We extracted the average soil carbon estimates from the soilGrid database down to 2m depth to compared against our reanalysis estimates.

# 3 Results and Discussion

## 3.1 Global sensitivity analysis and HPDA

The main purpose of the sensitivity analysis was to identify model parameters with a large influence on the behaviour of SIPNET output variables such as NEE. The estimated variability presented in Fig 5 averages the sensitivity of NEE to each parameter across multiple years and different sites, presenting just the parameters that met our 5% variance criteria. Because NEE is the difference between Gross Primary productivity (GPP) and total ecosystem respiration we expected that parameters that primarily regulate modeled GPP and respiration would be the most sensitive with respect to NEE. Soil organic matter (SOM) decomposition rate, which controls heterotrophic respiration, was the most sensitivity parameter for both plant functional types. Following SOM respiration rate, photosynthetic optimum temperature (psnTOpt), photosynthetic maximum capacity (Amax), growth respiration factor, and soil respiration Q10 were found to be sensitive in both conifer and grass PFT. Amax and psnTOpt directly contributes to the GPP calculation, while growth respiration factor contributes to autotrophic respiration and determines the fraction of GPP that is available for growth. The soil respiration Q10 and psnTOpt both demonstrate SIPNET's high sensitivity to temperature. In addition to shared parameters, the conifer PFT also showed a high sensitivity of GPP to vapor pressure deficit (VPD), reflected in the importance of the slope (dVPDSlope) and exponent (dVpdExp) in that equation, and to specific leaf area (SLA), which reflects how much new leaf area a plant can 'buy' for an given investment in leaf carbon. The grass PFT, on the other hand, showed a high sensitivity to immedEvapFrac, which is the proportion of incoming precipitation that is intercepted and re-evaporated rather than being allowed to enter the soil and become available for transpiration.

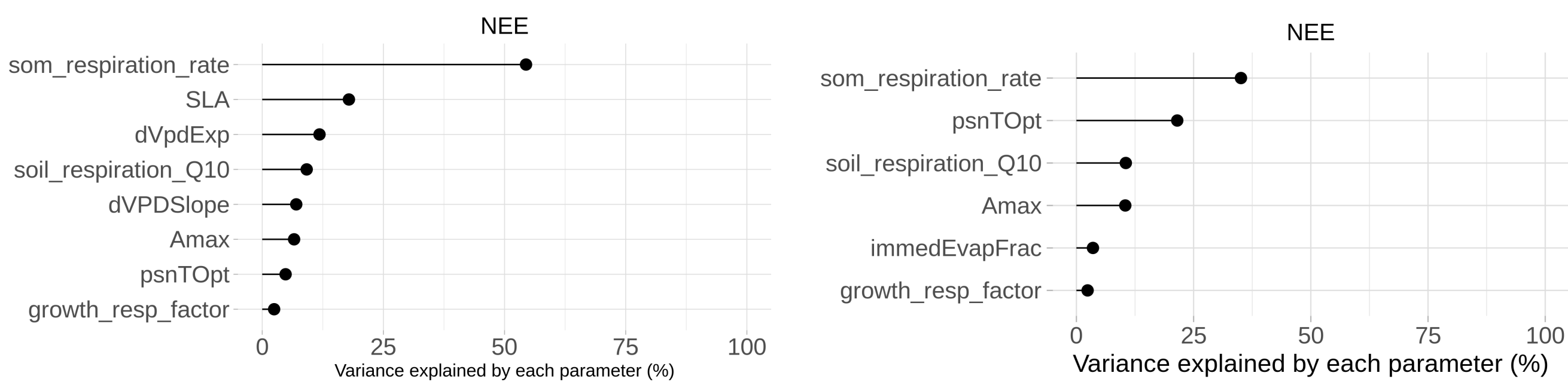

(a) Conifer PFT (b) Grass PFT

**Figure 5.** Global sensitivity analysis SIPNET model parameters to NEE prediction for different plant functional types

Sensitive parameters found in this study are similar to (Fer et al., 2018) where they estimated the sensitivity of SIPNET model parameters to NEE and latent heath flux for temperate deciduous PFT. Even though (Fer et al., 2018) used a local sensitivity analysis approach, compared the to GSA used in this study, SOM respiration rate, psnTOpt, soil respiration Q10 and slope of VPD-photosynthesis (dVPDSlope) were found sensitive in both studies. Specifically, the relative importance of

parameters in simulating NEE by SIPNET in grass PFT is similar to was found by (Fer et al., 2018), confirming the general behaviour of SIPNET model.

SIPNET is representative of a larger class of models, with simple representations of carbon pools and fluxes, which are efficient and thus useful for computationally intensive large scale data assimilation problems. After constraining the model against half hourly flux tower NEE, SIPNET showed considerable improvement (Fig 6). Nash Sutcliffe model efficiency showed an improvement for all years (2006-2011) and PFTS (a total of 134,000 combined observations for Grass PFT and 251,000 observation for Conifer) except for conifers in 2011 (Table 1).

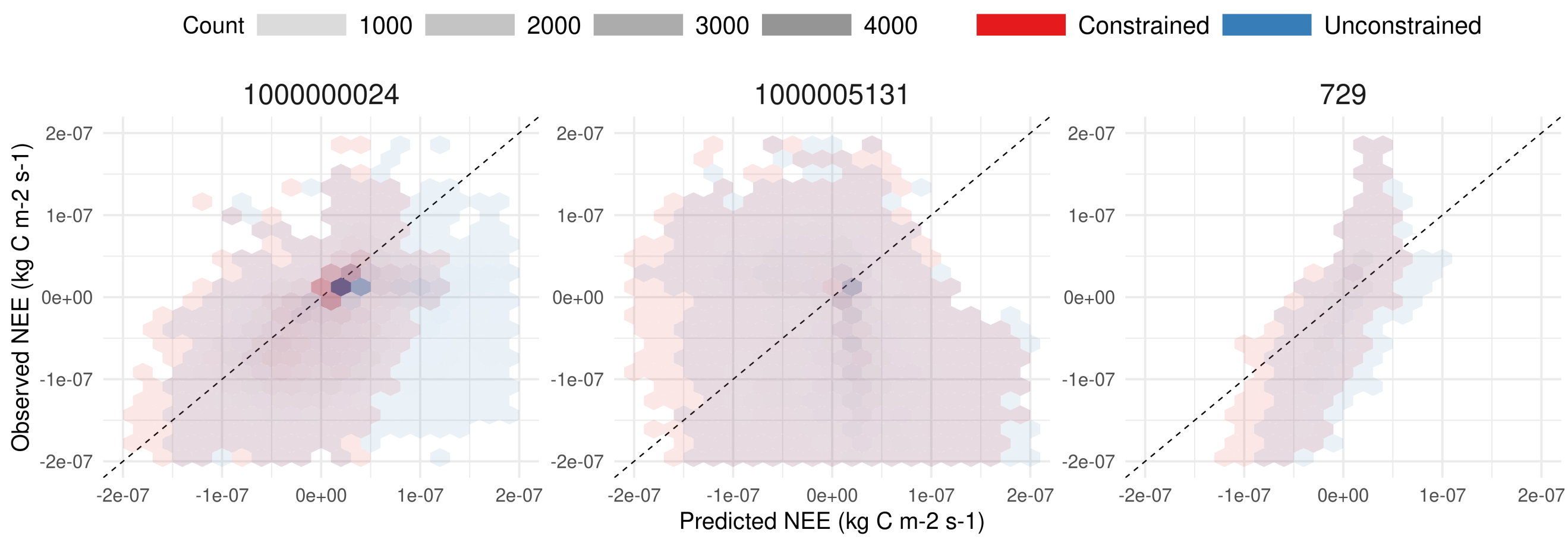

**Figure 6.** 1:1 plot comparison between observed NEE and simulated NEE for coniferous across all sites and years

Unconstrained simulations resulted in a large overestimation of NEE in almost all years and for all sites suggesting either overestimation of total photosynthesis or underestimation of total respiration. Constraining the photosynthesis and respiration parameters (Fig 5) improved the performance of the model, yet the overall underestimation behaviour in the SIPNET model (Table 1) even after calibration points to the inadequacies in the model structure to account for complex interactions among processes.

RMSE estimated in this study for Conifer ranged from 39 to 132 $\times 10^{-9}$ Kg C $m^{-2}$ $s^{-1}$, while Grass PFT ranged 68 to 174 $\times 10^{-9}$ Kg C $m^{-2}$ $s^{-1}$ across 2006 - 2011. (Fer et al., 2018) estimated a comparable 43 $\times 10^{-9}$ (Kg C $m^{-2}$ $s^{-1}$) RMSE for temperate deciduous PFT for two years of simulation and across 1 site. Both model assessment indices suggest that the SIPNET model is more constrained after calibration and ready for use within the state data assimilation routine.

### 3.2 State data assimilation

As mentioned before, SIPNET represents the terrestrial carbon budget by keeping track of three carbon pools, plant wood carbon, plant leaf carbon and soil carbon, and simulates the major carbon fluxes among these pools (allocation, turnover) and between these pools and the atmosphere (photosynthesis, autotrophic and heterotrophic respiration) (Fig 1). In our data

**Table 1.** NSE, RMSE and Percentage Bias (estimated as $\frac{sum(sim-obs)}{sum(obs)} \times 100$) estimated before and after parameter assimilation for the Grass and Coniferous plant functional types across all sites

| | | | $RMSE \times 10^{-9}$ | | NSE | | Percentage Bias | |
|---|---|---|---|---|---|---|---|---|
| Site ID | AmeriFluxID | PFT | Constrained | Unconstrained | Constrained | Unconstrained | Constrained | Unconstrained |
| 1000000024 | US-Fuf | Conifer | 47 | 162 | 0.06 | -10.1 | -97 | -429 |
| 1000005131 | US-Me2 | Conifer | 117 | 117 | -1.17 | -1.15 | -90 | -113 |
| 729 | CA-Ca3 | Conifer | 48 | 62 | 0.55 | 0.25 | -51 | -82 |
| 1000000139 | US-Br1 | Grass | 190 | 252 | 0.02 | -0.8 | -88 | -253 |
| 1000000141 | US-Br3 | Grass | 164 | 207 | 0.01 | -0.6 | -67 | -240 |
| 1000000142 | US-Bkg | Grass | 61 | 86 | -0.11 | -1.18 | -116 | -462 |

assimilation procedure, we sequentially assimilated above-ground biomass into the plant wood carbon and leaf area index into the plant leaf carbon on an annual time step, allowing soil carbon to be constrained based on the covariance estimated through the process model. Data assimilation was performed at all the sites at the same time where off-diagonal values in the process-model error covariance was adjusted according to Eq 9.

As an example of the general behaviour of the SDA, Fig 7 shows the reanalysis for a representative conifer site. The initial forecast starting from the BADM initial conditions has a wide posterior distribution and typically a relatively unconstrained mean forecast. The Analysis, on the other hand, converges quickly towards the observed AGB and remains thereafter as the compromise between the observation and the model forecast given their uncertainties. If no observation was available, for instance in the case of LAI from the beginning of the simulation until the year 2000 (Fig 7), the analysis step relied on the covariance estimated among the state variables, both with-in site and adjacent sites. The ability to take advantage of the covariances for updating a state variable with no observation allows the SDA to constrain variables, such as soil C, that lack direct, site-level observations at large scales for data assimilation. At the end of each time step, ensemble members were adjusted given the estimated $\mu_a$ and $P_a$ and used as an initial condition for the next time steps.

Starting in year 2000, we added a second observational constraint, MODIS LAI, and after which the Analysis LAI contracted considerably around the observations. The soil C pool reanalysis was variable depending on the observation error or uncertainty in the driving forces for different sites and did not show an obvious response to the addition of LAI as a constraint (Fig 7).

### 3.2.1 Flux validation and ensemble weighting

The knowledge gained through data assimilation about the state of the carbon pools at each site and year was then transferred post-hoc to the flux time-series over the preceding time interval by estimating the posterior probability of each ensemble member and using these probabilities to weight each ensemble member. At this stage, no additional observations were used to adjust the model estimates of different carbon fluxes such as NEE (Fig 8). For example in the Black Hills Ameriflux site (Fig 8), the NEE forecast by SIPNET, which already accounts for the initial condition constraint from the previous SDA step and the

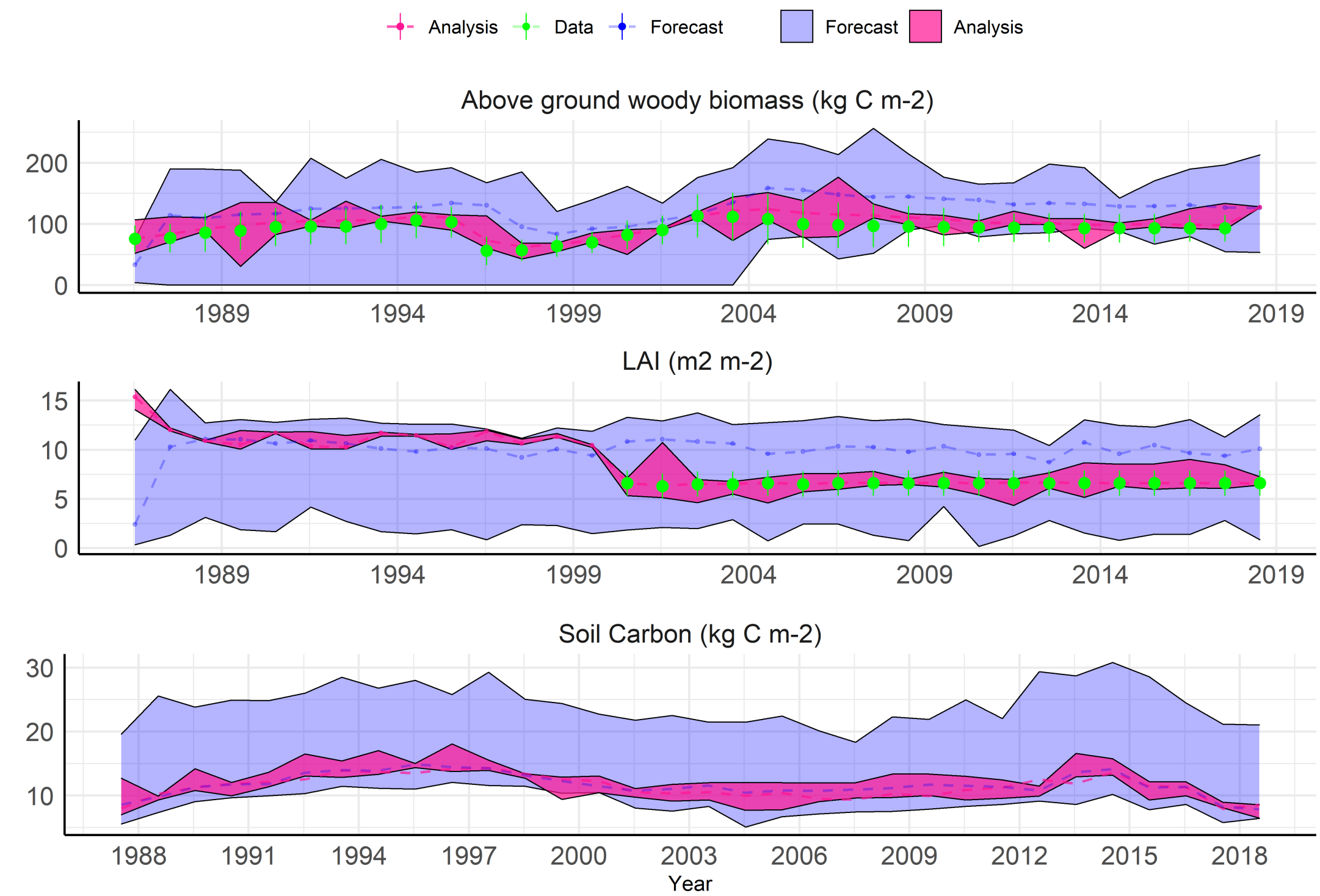

**Figure 7.** Site-level example of sequential data assimilation at Eastern Temperate Forests ecoregion for a Coniferous site. (Longitude=-89.226 Latitude=33.21)

parameter constraints from the HPDA, closely followed the observed value. However, adjustment of the model estimates using post-hoc weights further improved the predictability of NEE. The effect is not huge, as the current SDA constraint reflects n=2 observations per year, however the post adjustment RMSE was reduced by 7.8% on average for the time period between 2004-2009. Adjusting NEE estimates solely based on the weights estimated through data assimilation helped to further close the gap between our forecast and observed NEE even after parameter calibration. The improvements provided by assimilating just two observations per year points to the obvious future directions of increasing both the number of different data constraints and the frequency at which data are assimilated.

Examining the weights assigned to each ensemble member and the flux forecasts associated with them (Fig 8 top panels) shows two useful patterns. First, there are ensemble members that get consistently low weight, and this includes most of the ensemble members that predict unusual flux time-series (e.g. ensemble members 20, 16, and 11). These ensemble members are likely associated with parameter combinations that are not representative of this site (and potentially across sites) and points to a future opportunity to combine simultaneous state estimation and hierarchical parameter constraints (Dokoohaki et al., 2021). This could be done by including the parameters in the analysis and ensemble adjustment, effectively nudging parameters toward more supported values, or through a parameter filtering and resampling, as is done in a particle filter (Dietze, 2017). The second

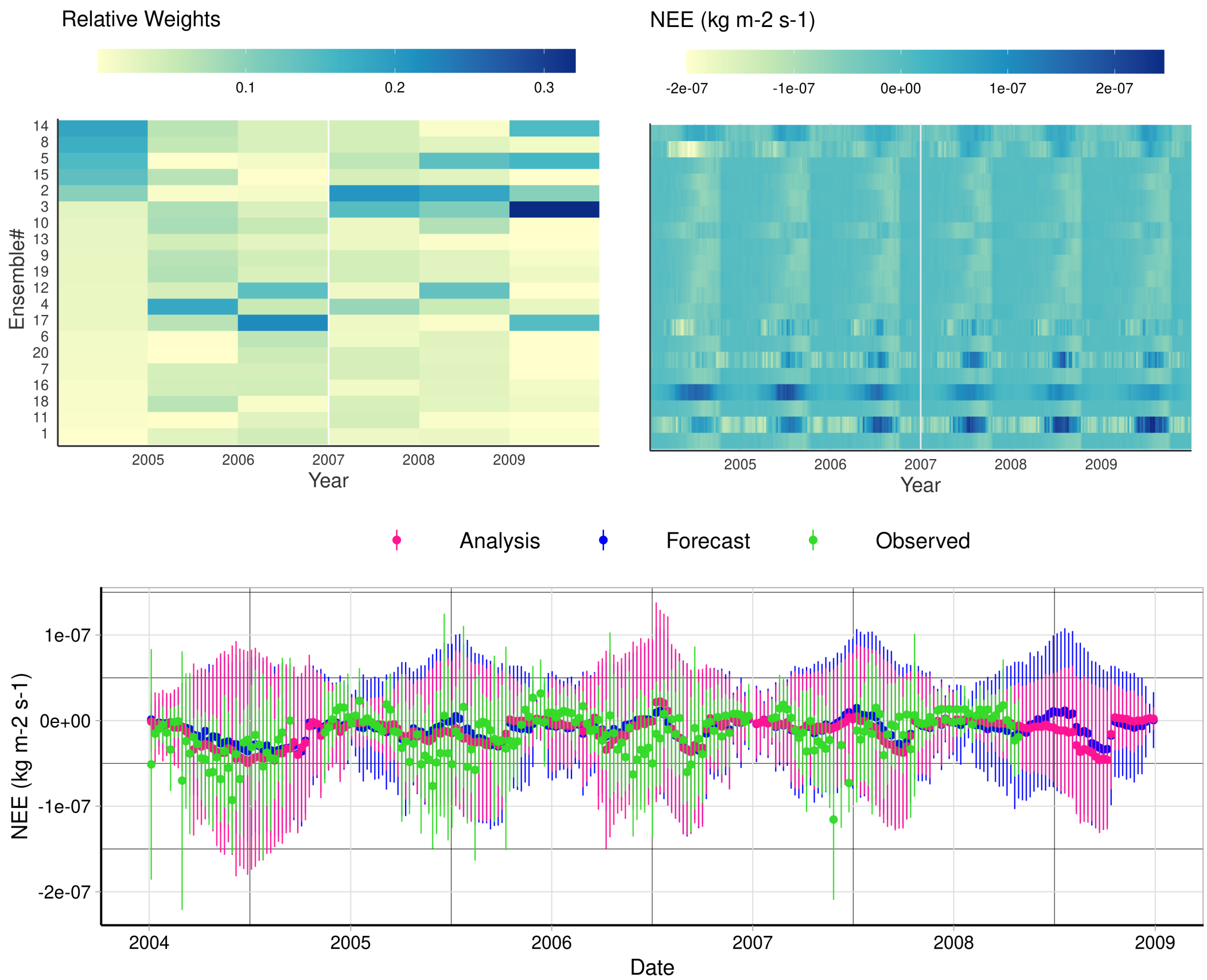

**Figure 8.** Top left: Weekly NEE predictions for all ensemble members at the Blackhill site; Top right: Relative weight estimated for each ensemble member using Eq 10 sorted by cumulative weight from highest (top) to lowest (bottom); Bottom: Weekly average adjusted NEE (Analysis) compared to the observed measurements

pattern that is apparent in the ensemble weights is that the same ensemble members are not performing the same every year. For example, an ensemble member may perform well one year, only to get very low weight the very next year, but then perform well again a few years later (i.e., ensemble members 2 & 5). To the extent that this reflects variations in parameters, not just initial condition and driver variability, this also suggests that the 'best' parameters may vary through time. An important future direction to follow from this is to perform a more in-depth analysis of how model parameter weights are varying in both space and time to better understand what parameters are most variable, as this points to processes that are unaccounted for in the underlying models (e.g. if SOM respiration rate varies spatiotemporally it suggests the need to add additional terms to account for this variability). Furthermore, examining the patterns to this variability in space and time can provide important clues about the underlying factors driving variability. Finally, we would need to extend our hierarchical parameter constraint to account for spatiotemporal variability in parameters, as opposed to the current approach that only captures spatial variability.

It is important to note that the patterns of ensemble weights in Fig 8 reflect the information contribution from the MODIS LAI and LandTrendr AGB constraints, not the information available from the NEE observations themselves. For future applications of this approach, there are other options available to assimilate observed carbon, water and energy fluxes. The simplest would be to assimilate the cumulative fluxes that have occurred over the time since the previous analysis. We have implemented this approach as part of our site-scale near-term flux forecast (Helgeson et al *in prep*), and it introduces the challenge of having to gap-fill the observed fluxes to be able to integrate them. Gap-filling introduces uncertainties and causes the assimilation to be, in part, a model-model comparison (process-model vs gap-filling algorithm) instead of a model-data comparison. Assimilating cumulative flux also results in a loss of temporal information and statistical power. Alternatively we could calculate the Likelihood of the observed fluxes for each ensemble member and include this as a weight when calculating the forecast mean and covariance(i.e., $\mu_f$ and $P_f$). This approach is similar to a particle filter and to ensemble variational data assimilation (Pinnington et al., 2020). For either approach, there are important questions to be explored about the spatial range at which point-level flux observations provide meaningful constraint on flux reanalysis across landscapes and regions.

#### 3.2.2 Soil C Assessment

The degree to which our data assimilation algorithm is able to constrain the soil C pool, which is not observed directly but inferred from the other observations, is related to the quality of AGB and LAI observations at each site, the strength of the covariances between the plant and soil pools, and the overall accuracy of the model. A comparison between the distribution of soil C estimated in this study against the soilGrids database shows slightly higher mean soil C estimates in this study (13.11 KgCm2 VS 8.77) but comparable variability across all plant functional types (Fig 9). The median uncertainty in soil C reanalysis over all sites is between 15-20 Kg C $m^{-2}$, and this uncertainty did not change by adding LAI as a new data constraint. Our reanalysis did reasonably well at predicting SOC for the conifer PFT but was consistently higher than corresponding soilGrids estimates across all other PFTs. Arid grassland showed the largest overestimation with 153% bias compared to the SoilGrid dataset; whereas, the deciduous PFT showed the lowest deviation with only 14 % bias. Furthermore, The largest agreement (Index of Agreement (Willmott, 1981) $\sim 0.7$) was also found in conifer and deciduous PFTs between the two datasets and lowest agreement was found to be 0.28 for arid grassland.

In rare occasions (i.e., 2 sites in arid and 3 sites in mesic grassland) in this study, we produced estimates of soil C which are not compatible with estimates from other data products such as soilGrids. This divergence might be attributed to instability in our statistical model in sites with sparse observations. Alternatively, the mismatch between how soil C pool is defined (i.e., labile versus recalcitrant) between different machine learning and process-based models may complicate the comparison between these data products. It is important to note that this assessment of the performance of our reanalysis does not constitute a true validation because the SoilGrids product is itself a model. But, given that these two studies showed comparable means of soil C pool across ∼500 sites through two different methods with different philosophies, our soil carbon data products can be regarded as a new instrument that allows us to enhance the observational information and to derive higher-level products. In the future, as the spatial resolution of our reanalysis increases, it will be important to perform a more detailed assessment

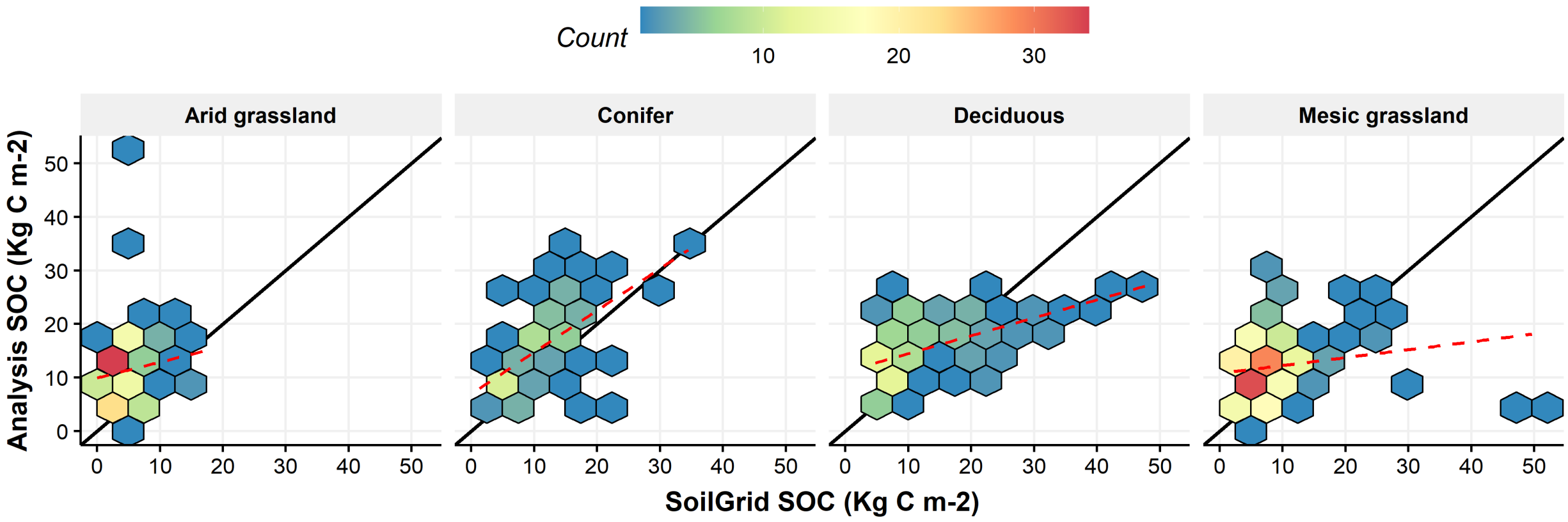

**Figure 9.** Soil Organic Carbon analysis comparison to soilGrids database for different plant functional types. Solid back line represent the one to one line and dashed red line is the linear regression line between reanalysis SOC in this study and soilGrid SOC.

against a wider array of detailed site-level soil data, such as the International Soil Carbon Network's database (Lawrence et al., 2020), the global soil respiration database (Blackard et al., 2008) and the National Ecological Observatory Network.

### 3.2.3 Forecast Uncertainty

One of the notable patterns in our proof-of-concept reanalysis was the tendency for the forecast uncertainty, $P_f$, to be fairly large despite finding that our reanalysis posteriors had fairly high precision. In part, this is occurring because this system is propagating a much broader set of uncertainties than previous studies. Variability in model forecasts can be attributed to the uncertainties in meteorological drivers, model parameters, initial conditions (i.e. analysis posteriors), and the unexplained process error (Dietze, 2017).

We did not find substantial shrinkage in average forecast variability for soil carbon and LAI pools after including the LAI observations into our data assimilation routine (Fig 10). In other words, adding a second data constraint did not result in proportional reduction in forecast uncertainty for LAI and soil carbon. However, we noticed a slight reduction in forecast uncertainty for above ground woody biomass after addition of the second data constraint. Large observation error in MODIS LAI and the inconsistency in scale between the LANDSAT and MODIS resolution could be a few reasons for this observation. Similar findings were reported by (sch), where their multi-stream carbon DA exercises found that a single data stream fit almost as well as multi-data stream assimilation, and by (Castro-Morales et al., 2019), where they found the largest portion of information came from the first decade of data assimilation, similar to what was found in this study. Altogether, this suggests that initial condition uncertainty is not the dominant driver of forecast uncertainty in the one-step-ahead predictions that occur within the forecast-analysis cycle. Previous site-scale assimilation within the same framework, but with a different model (LINKAGES), found process error, Q, to be a dominant source of forecast uncertainty (Raiho et al., 2020).

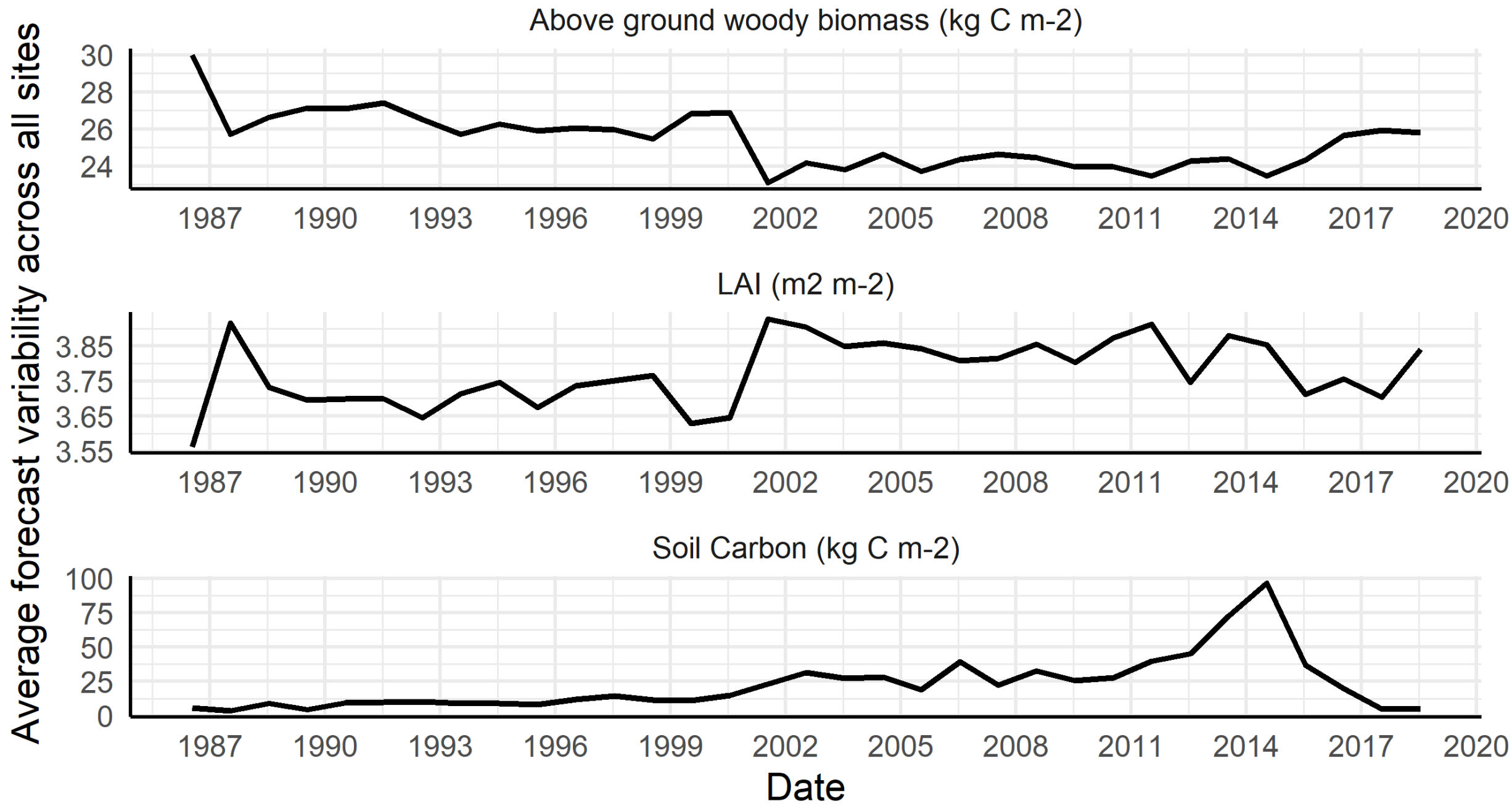

**Figure 10.** Average forecast variability (sd) across all sites and all pools at each time step.

455 Overall, an important future direction for this work is to perform a detailed uncertainty analysis of the regional-scale reanalysis. Similar to Raiho et al. (2020), this should partition the uncertainties among the four major drivers (initial conditions, drivers, parameters, process error). The dominant uncertainties should then be further partitioned to better understand the contributions of individual processes and parameters (Dietze, 2017, 2014). This feedback will be important for further refining the system, for example to prioritizing work among further calibration, additional SDA data constraints, refinement of the mechanistic model, and refinement of the process error covariance model. It may also identify the contributions of uncertainties that are less reducible, such as errors in drivers and stochastic disturbances.

#### 3.2.4 Future Directions

While SDA is a valuable tool for producing reanalysis products and improving forecasts, it cannot improve how well the process model represents the system and how often we assimilate new observations. While a simple model like SIPNET presents an opportunity for testing new statistical models for data assimilation and expanding to larger scales, it lacks mechanistic detail around features such as disturbance, phenology, plant carbon partitioning, vegetation composition, structure, and demography, soil and plant hydraulics, and soil carbon dynamics. As a future direction, we will be working towards employing a more advanced vegetation demographic model (Fisher et al., 2018), such as ED2 (Medvigy et al., 2009; Moorcroft et al., 2001), into our data assimilation scheme.
In addition to increasing model sophistication, there are clear arguments for increasing spatial resolution, temporal resolution, number of ensemble members, and number of data constraints. However, increasing all of these simultaneously will greatly increase computational demands, especially if we also employ more advanced (and slower) models. Prioritizing which dimensions to focus on first will depend on a number of factors. First, we need to perform more formal assessments of the computational costs of each of these dimensions, and the accompanying reductions in uncertainties that each affords. The answers to these questions will also depend on the needs of different end-users. For example, in our companion paper (Morrison et al *in prep*), we develop an additional statistical upscaling layer on top of our data assimilation model, providing a first pass at generating an overall, synthetic CONUS-scale carbon budget. We then ask how reducing spatial resolution affects uncertainties at CONUS-scale. Furthermore, an additional advantage of our use of an irregular spatial grid is that further spatial refinement does not need to be distributed uniformly, but can be concentrated where needed to reduce uncertainty or improve understanding. This could be the places contributing the most uncertainty to our overall CONUS C budget, but it also could be places of particular concern to end users (e.g. policymakers, land managers, scientists trying to better understand C cycle variability). An analogous approach can also be taken in time, concentrating sampling on the seasons needing the most constraint (e.g., phenological transitions). Additional analyses are needed to identify and prioritize these spatial and temporal domains.

In terms of increasing the number of data constraints, there are a large number of opportunities available to those interested in carbon cycle reanalysis and forecasting (Baatz et al., 2018). As mentioned earlier, tower fluxes provide a way of anchoring reanalysis estimates of fluxes across broader scales. In addition, there is a clear need to constrain soil C pools and fluxes using a combination of detailed site-scale data and more broad spatial data products. One challenge with assimilating soil C is that most spatial products do not have specific dates associated with them, but rather aggregate data across a broad time range. To assimilate products like this, we will leverage an approach we developed for assimilating fossil pollen data over long time scales (Raiho, 2019) that spreads information over time by inflating the uncertainty such that the effective sample size (and thus information content) is preserved. For CONUS-specifically the USFS Forest Inventory and Analysis also provides additional constraints on biomass and soil C (Blackard et al., 2008).

Beyond field-based measurements, there is also a wealth of remotely-sensed data that provide additional constraints. For example, there are a wide range of optical multispectral platforms that could constrain LAI, vegetation type (PFT), and land use change both deeper in the past (e.g., AVHRR, Landsat) and moving forward (e.g., VIIRS, Sentinel-2). Active sensors like LIDAR (i.e., GEDI, ICESat) and radar (i.e., PalSAR, BIOMASS, NISAR) also provide additional constraints on vegetation height and aboveground biomass. In terms of fluxes, solar-induced fluorescence (i.e., OCO-2, OCO-3, GOME-2, TROPOMI) provides a constraint on GPP, while the same sensors provide column-averaged CO2 estimates that potentially provide a flux constraint on NEE. CO2 concentrations are usually related to land fluxes via complex atmospheric inversions (Ciais et al., 2010), and here, terrestrial carbon reanalyses provide an opportunity to formally reconcile top-down inversion and bottom-up inventory estimates of carbon fluxes and sink attribution. Specifically, atmospheric inversions require land priors, which are typically fairly uninformative. Land carbon reanalysis products provide a new opportunity to provide much more informative priors to atmospheric inversions, which provide data constrained estimates of not just means and variances, but also the covariances across spatial and temporal scales and among pools and fluxes.

In addition to direct carbon constraints, there are also a number of remote data products that provide indirect constraints on carbon pools and fluxes. In particular, carbon and water are tightly coupled in many ecosystems (Schlesinger et al., 2016), such as the trade-offs between photosynthetic C update and water loss through transpiration that is measurable via eddy-covariance latent heat flux. Microwave remote sensing (e.g., SMAP, SMOS, Sentinel-1C) provides estimates of both soil moisture and vegetation optical depth, the latter providing information on both vegetation biomass and moisture status (Konings et al., 2017, 2019). Similarly, tower measurements of sensible heat flux and satellite measurements of land surface temperature (e.g., ECOSTRESS) provide information about plant stress (Hook and Hulley, 2019). Finally, imaging spectroscopy (e.g., AVIRIS, DESIS, SBG) provides detailed information about composition, stress, and canopy properties, including the ability to constrain many of the parameters critical to terrestrial ecosystem models (e.g., SLA, Vcmax, leaf N) (Roberts et al., 1997; Shiklomanov et al., 2021).

The localization scheme employed in this study was adopted from climate forecasting research (Reich and Cotter, 2015), where the system they try to simulate is similar in neighboring cells or simulation blocks. However, a measure of physical distance alone may not be adequate for data assimilation in ecological systems, as two adjacent modeling blocks could represent completely different ecological systems such as different plant functional types, forest stand types etc. We found that this may substantially affect the process error in the absence of direct observations. This reveals a need and also an opportunity for further exploration of different and more efficient localization schemes in ecological forecasting. Understanding the ability to scale information across space is particularly important for extending this proof-of-concept to other regions of the world that are less data dense.

## 4 Conclusions

In this study, we presented a proof-of-concept for a new, synthetic “reanalysis" data product that harmonizes the different components of carbon cycle across the contiguous United States. Overall, our system, which is open source and extensible to other models, successfully scaled up the new TWEnF assimilation algorithm and PEcAn workflows to provide an unprecedented level of error accounting, including the first estimates of model process error covariance at a national scale. To do so, we developed an efficient approach for assimilating data on an irregular grid and an ensemble weighting approach for updating high temporal resolution fluxes. Before running the assimilation, we successfully constrained SIPNET model parameters in a hierarchical Bayesian parameter data assimilation framework using NEE observations from 6 sites for two PFTs. Next, MODIS LAI and LandTrendr above ground biomass were sequentially assimilated into the SIPNET model across a network of 493 sites to constrain estimates of carbon pools (leaf, wood and soil C) and fluxes (GPP, Ra, Rh, NEE). The comparison between our SOC estimates and the soilGrids data product showed that we have been able to successfully constrain the SOC pools, while comparisons to eddy covariance demonstrated that were able to constrain net ecosystem exchange. Ultimately, carbon cycle reanalysis should be scalable to a global extent, providing a uniform synthetic platform for carbon monitoring, reporting, and verification (MRV) and accelerating terrestrial carbon cycle research.

*Code and data availability.* All the code developed for this study can be found at https://doi.org/10.5281/zenodo.5557914, last access: 10 Oct 2021. Leaf C, wood C, soil C, NEE, GPP, auto and heterotrophic respiration data products can also be found using the following link: https://osf.io/efcv5/?view_only=f2eeea87a6504abbae81164efd2b481c

*Author contributions.* H. Dokoohaki was responsible for developing and running the regional scale data assimilation codes, and conducting the study; B.Morrison was responsible for preparing the observe data used in the SDA and the site selection procedure; A. Raiho was responsible for developing the site-scale data assimilation code; S. Serbin, A. Andrews, and M. Dietze were responsible for developing the initial idea and supervising the study. H. Dokoohaki and M. Dietze were responsible for writing the manuscript and all other authors contributed to revising the manuscript

*Competing interests.* The authors declare that they have no conflict of interest.

*Acknowledgements.* This research was supported by NASA CMS under award number 80NSSC17K0711. The PEcAn project is supported by the NSF (ABI no. 1062547, ABI no. 1458021, DIBBS no. 1261582), NASA Terrestrial Ecosystems, the Energy Biosciences Institute, and an Amazon AWS education grant. B.D.M and S.P.S were also partially supported by by the United States Department of Energy contract No. DE-SC0012704 to Brookhaven National Laboratory.

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

## Appendix A: Localization

Localization has been found to be one of the most important determining factors in the success of high-dimensional data assimilation problems specially EnKF (Farchi and Bocquet, 2019). Localization is often applied to either the observation or forecast error covariance matrix with the goal of reducing or removing the spurious correlations that might be generated due to low sample/ensemble size. In this study, we used a local support localization method (Eq.9) adopted from Kalnay and Li (2010), where the localization function returns non-zero values for small distances (local region) and zero for elsewhere. In order to explore the effect of the scaling factor in Eq.9, we setup an experimental data assimilation simulation with all 49 Ameriflux sites and a series of scaling factors (ranging from 0 to 900km) for the time period between 1989 - 2000. In this

experiment, we were interested to find a scaling factor such that it would produce the lowest spatial autocorrelation in the residuals after the data assimilation. Given our simulation setup, we found that increasing the scaling factor from 0 to 900 Km would initially decrease the spatial auto correlations (Fig A1) from $\nu = 0$ to $\nu = 500$km and then the autocorrelation would plateau and slightly increase form $\nu$ greater than 500 Km. Using the $\nu = 500km$ with lowest spatial autocorrelation, in one hand ensures that our data assimilation algorithm takes advantage of local correlation among similar sites while, on the other hand it ignores/removes the spurious correlation estimated at larger distances.

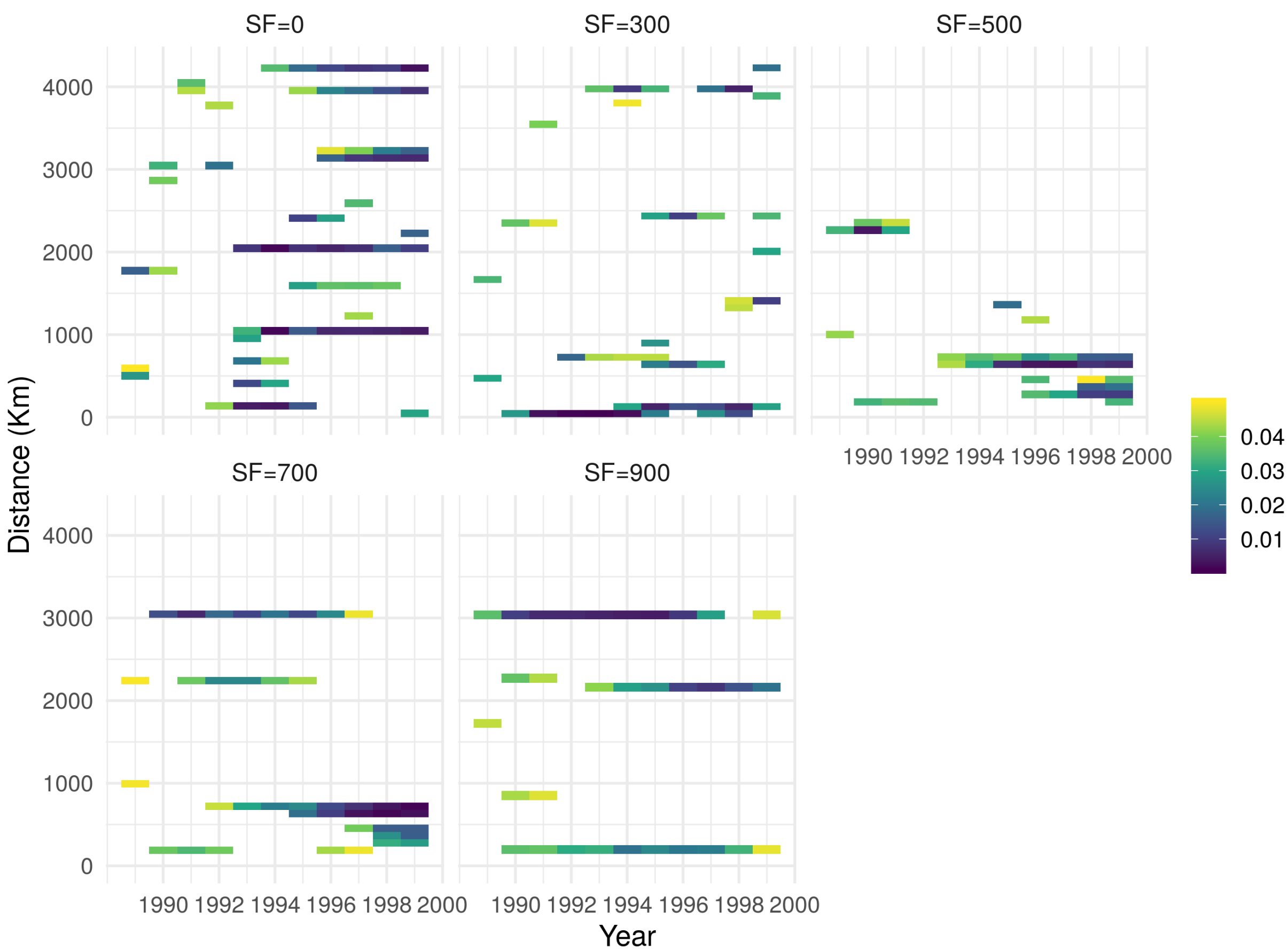

**Figure A1.** Significant spatial autocorrelation (p-value < 0.05) found for different values of scaling factors ($\nu$) in Eq.9 across a range of distances and years (1989-2000).