# Peer review of "Development of an open source regional data assimilation system in PEcAn v. 1.7.2. Application to carbon cycle reanalysis across the contiguous U.S using SIPNET."

_Geoscientific Model Development, 2021_

## Author Response (AR1)

**RC1**

**Major comments**
Please refer and evaluate previous studies. There are several articles you might want to
Consider.

*1- The target temporal range (1986-2019) is relatively long. You may need to consider the effects of succession and/or landuse change in this timescale.*

*Succession:*
*We agree that the impacts of succession would be a useful avenue for future exploration using cohort-based models that are better able to track species competition and the resulting changes in vegetation composition and structure (e.g. Fisher et al. 2018 GCB DOI:10.1111/gcb.13910). Indeed, we have already applied this same assimilation approach at the site-scale over longer time periods using the LINKAGES model and a combination of forest inventory data and tree-rings as constraints (Raiho el al. 2020 https://doi.org/10.1101/2020.05.05.079871) but for the "proof-of-concept" this paper was intended judged that the added compute and complexity was not needed. Unlike the Raiho paper, initial data constraints considered here lack the same PFT-level resolution, and applying data assimilation to the sum of the AGB pools in order to constrain the biomass of each individual PFT is non-trivial and likely statistically nonidentifiable. Furthermore, the cohort-based model we already have coupled to our system (LINKAGES) only works for forests and we have only calibrated it to the northeast and north central US. In the future we hope to extend our system to cohort-based models that are capable of continental-scale application (e.g. ED2, FATES, LPJ-Guess) but that is a significant investment that is far beyond the scope of this paper. Overall, the advantages of these alternative models are already noted in our Future Directions so we have not added text to the paper.*

*It is also worth noting that there is a lot of evidence that PNET/SIPNET specifically, and pool-based models in general, can capture carbon pool trajectories over this time scale. Indeed, almost all IPCC / CMIP projections to 2100 and beyond are done using Earth System Models that employ pool-based land models. Furthermore one of the advantages of data assimilation over simple forward model simulation is that it will "nudge" projections back on track when there are errors in the model.*

*Land use change:*
*Yes, large disturbances and changes in PFT composition are possible sources of error/bias. How and why this occurs within data assimilation systems, and new data assimilation algorithms that address this problem using Multinomial-Tobit mixture models, were the subject of recent conference presentations by co-author Dietze (ESA 2021, AGU 2021) and are the subject of a paper we have in prep. Explaining this issue in detail, and correcting for it, is clearly beyond the scope of the current paper, but we have added the following to the Future Directions:*
*"It is also worth noting that conventional data assimilation approaches do not do a great job at capturing the discrete nature of large disturbances or changes in land cover, leading to biases in pool and flux estimates both at the time of the event and over the next few years as these algorithms "nudge" pools back on track. While these errors are likely very local in space and asynchronous compared to the CONUS scale being considered, we are currently exploring alternative algorithms (Dietze et al. in prep) and hope to incorporate improved algorithms in the next generation of this system.*

*2- Please explain why you can say that the very simple model SIPNET (a "big-leaf" model) was able to reproduce the terrestrial ecosystem dynamics in this range.*

*SIPNET has consistently shown its capacity in estimating different components of the terrestrial carbon cycle across numerous studies. For example, Braswell et al., 2005 showed that when SIPNET is optimized using a Bayesian approach it can effectively simulate NEE at both diurnal and seasonal temporal scales. Fer et al., 2018 showed another successful application of the SIPNET model in simulating NEE, Latent heat flux (LE) and soil respiration at half hourly time steps. Finally Sacks et al., 2006, employed SIPNET model to extract process-level information from eddy covariance data and they reported that using SIPNET they were able to learn about the primary controls over NEE at an evergreen forest in the Colorado Rocky Mountains, and in particular the respiration component of NEE.*

*Sacks, W.J., Schimel, D.S., Monson, R.K. and Braswell, B.H., 2006. Model‑data synthesis of diurnal and seasonal CO2 fluxes at Niwot Ridge, Colorado. Global Change Biology, 12(2), pp.240-259.*

**3- Arakida et al. (2017) is an example of data assimilation of hard-core individual-based model, with explicit representation of vegetation dynamics such as competition and succession. Please explain strengths and weaknesses of your approach against this preceding work. You may need to rewrite the Future Directions according to this information.**
*Thank you for the comment. We have included more studies in our background review of recent data assimilation works in the introduction section Line 84- 87 and 94-99. Arakida et al. (2017) was also cited and added to this discussion. Below is a compare/contrast of Arakida et al., (2017) with the this study:*

*Arakida et al., (2017) used a Particle Filter (PF) to be able to relax EnKF assumptions when assimilating LAI into a land model. While the PF is even more flexible than the TWEnF, it requires a much larger ensemble size (>100x), and this analysis only considered the uncertainty in model parameters, which were drawn from an uninformative distribution, and did not include driver uncertainty or process error.*

**4- Ise et al. (2018) optimized a simple TBM using particle filter, a primitive approach of DA. They used a brutal power of a supercomputer to optimize the model on big spatiotemporal data. It might be beneficial to read and compare.**
*Thank you for introducing the paper. Ise et al., (2018) is solely focused on parameter data assimilation rather than state data assimilation. They optimize nine model parameters with PF using LAI observations. PF is capable of relaxing some of the EnKF assumptions shared with theTWEnF SDA, but as reviewers noted that approach is far less computationally efficient than the approaches employed in this work.*

**5- Please summarize the computational environment of this study. What was the computational burden (number of sites x number of timesteps)? How long did the computation take? Your approach using an improved TWEnF was really needed for this Exercise?**
*Thank you for the question. We added information regarding the SDA run time in line #296-299, while more information regarding the number of sites x number of timesteps was added in line #285-287. Lastly, as it was mentioned in the material and methods section and under state data assimilation, the TWEnF was employed  because it allows for relaxing assumptions relative to other Kalman-style filters, while requiring a much lower ensemble size than PF.*

**6- Please summarize the parameters optimized by the SDA.**
*No parameters were optimized in the SDA. SDA was run with the joint posterior parameter samples from the PDA.*

*Minor comments*

**7 - p.2 l.47 I think there are a few terrestrial ecosystem models that explicitly consider the uncertainties in model parameters. Please try to find them.**

*We agree with reviewers' comments that while still rare, the consideration of parameter uncertainty in terrestrial ecosystem models in general, and terrestrial ecosystem data assimilation in particular, is increasing. . Therefore, we performed an additional review of the literature and amended the introduction. From the list of newly reviewed papers, Ise et al., (2018) might be considered one, even though that study is solely dedicated to parameter estimation and not state data assimilation.*

**8 - p.13 l.333 You just found tendencies that was already described by a previous work. Please explain why this fact strengthens your study.**

Here we demonstrate that the most uncertain parameters in the SIPNET model for the conifer and grass PFTs are similar to what was found for the deciduous tree PFT using a slightly different approach. This finding was not unexpected, nor did we claim it was unexpected, as our prior work with other models demonstrated that dominant uncertainties are often consistent across PFTs and locations (Dietze et al 2014 JGR-B doi:10.1002/ 2013JG002392). Nonetheless the dominant uncertainties for new PFTs in this model still needed to be identified and documented prior to calibration (PDA) to ensure that we are including the correct set of parameters in the calibration.

**RC2**

**Currently, the results as presented do not really represent a "reanalysis across the contiguous US". Indeed, there is no discussion of regional/continental scale biomass or flux estimates at all, which is surprising given the article title and abstract. It does seem like these should be amended to reflect the end point of this current study more accurately.**

*We specifically submitted this paper under "Development and technical paper" rather than as a research paper to demonstrate a proof of concept carbon data assimilation system that as the referee agrees is "a complex, technological feat and a successful development of impressive workflow capabilities". However, we agree that the title and parts of the abstract needs to be modified to reflect the technical nature of this manuscript and therefore we changed the title to :*

> *"Development of an open source regional data assimilation system in PEcAn v. 1.7.2. Application to carbon cycle reanalysis across the contiguous U.S using SIPNET."*

**The authors suggest that it is technically very challenging to produce a detailed land surface reanalysis using a complex carbon cycle model contiguously at regional or continental scales. Nonetheless, this is now routinely done by multiple different groups (e.g. Albergel et al., 2017, 2010; Bacour et al., 2015; Boussetta et al., 2015; Demarty et al., 2007; Kumar et al., 2019; Ling et al., 2019; Raczka et al., 2021) and this work needs to be acknowledged, and critiqued if applicable.**

*Thank you for the comment and introducing the above papers. We closely reviewed all and incorporated relevant works (studies that are only focused on parameter optimization may not be totally relevant to this work) in our previous discussion of limitation in earlier works section line #80-97 . We agree that there are other systems capable of regional-scale terrestrial C data assimilation, however this does not mean that such systems are not technically challenging to engineer, or that our system does not provide unique advantages. Most studies, including the ones mentioned above, are limited by either their spatial extent (e.g. single site), the set of uncertainties that are propagated (e.g. missing some combination of process, driver, or parameter uncertainty), number of data constraints, demonstrated ability to support multiple models, or the assumptions of their statistical model. Data assimilation might*

*be routinely done at a single site but as the number of sites increases new opportunities and challenges arise (e.g. how to leverage spatiotemporal cov between the sites) that makes analysis such as the current study or Raczka et al., 2021 unique in their ability to overcome those limitations.*

**Given that it will be a companion paper in preparation that will describe a regional/continental scale analysis, the question then becomes what is the contribution of this paper beyond the excellent works of Fer et al 2018 and Raiho et al 2020, which thoroughly describe the methodologies employed here?**
This work focuses on unifying the approaches already developed by Fer et al., 2018 and Raiho et al 2020 and expanding on this framework by developing new components with the most significant being the significantly larger scale (expanded input processing pipelines, consideration of cross-site covariances and the implementation and calibration of a localization approach), a novel stratified site selection procedure, and adding a new model and additional data constraints. Altogether we offer an open source, unified workflow that has been coupled to multiple models and leverages well established methodologies for a regional carbon cycle reanalysis.

**This could be a detailed technical assessment of skill, or otherwise, of the model and data assimilation system. The results and discussion does begin to do this, but should contain a range of additional, rigorous metrics that are routinely used describing the performance of both parameter estimation and data assimilation – for example prior ranges and posterior distributions on parameters, the number of observations available for assimilation, the number rejected due to QC, measures of bias, histograms of residuals, etc. These, and others, are required to be able to judge the performance of the system and move beyond qualitative description**.
Thank you for the comment. In addition to the Table1 which provides a detailed assessment of model performance of the parameter estimation, we added posterior and prior density plots of both PFTs to the Supplementary materials Fig A1. Furthermore, the total number of observations used in SDA was added to line 285-287. Residual density plot for sites in Fig 6 was also added to the supplementary materials.

**For example, suggesting that "SIPNET showed considerable improvement" (line 341) after optimization seems quite meaningless, not least because at some sites model performance doesn't improve at all. What is the cause of the marked variability in improvement between sites?** *The spread in Figure 6 and the remaining biases after optimization in Table 1 seem to actually call into question whether this model is able to produce meaningful results at all when trying to represent a "PFT", rather than a single site location for which it was originally designed. This needs to be thoroughly discussed. Suggest that "Freerun" or "Open-loop" is used rather than "Forecast" in Figures 7 and 8 to make it clear that this has not been constrained by the DA system from an initial condition.*
*As it was discussed earlier in the manuscript, continental scale carbon cycle analysis requires a tradeoff between computational efficiency (model simplicity) and how well we can capture complex ecological processes such as the carbon cycle. In this study we explored the application of a simple process model (SIPNET stands for **Simplified** Photosynthesis and Evapotranspiration model) for capturing highly dynamic half **hourly** Net EcoSystem Exchange (NEE) across different ecosystems with different environments. In section 2.1, we explain the assumptions made in developing the SIPNET model and the tradeoff between its computational efficiency and model complexity. Given a **simple 3 carbon pool model** (Fig 1), we simulate NEE at hourly time scale and we believe given the simplicity of the model, we found **considerable** improvement (lower RMSE (in 5 out of 6 sites), lower Bias (In all sites) and more favorable NSE (In 5 out of 6 sites) (Table 1)) **across 251,000 observations** from 5 different years and 6 sites. Of course this improvement was not uniform across all the years and all the sites and that goes back to the tradeoff that a simple model is not able to capture every process in the carbon cycle at a half hourly time step, **however it captures the overall trend**. Furthermore, numerous model-intercomparision projects have shown that ALL land models struggle with capturing site-to-site variability, even models with much more complex structures than SIPNET (e.g. Schwalm et al 2010). Moreover, one of the main*

*advantages of performing state data assimilation is that it allows for reducing model simulation error that can arise from model structure (Raczka et al., 2021).*

*Schwalm C, C Williams, K Schaefer, R Anderson, M Arain, I Baker, A Barr, TA Black, G Chen, J Chen, P Ciais, K Davis, A Desai, M Dietze, D Dragoni, M Fischer, L Flanagan, R Grant, L Gu, D Hollinger, RC Izaurralde, C Kucharik, P Lafleur, B Law, L Li, Z Li, S Liu, E Lokupitiya, Y Luo, S Ma, H Margolis, R Matamala, H McCaughey, R Monson, W Oechel, C Peng, B Poulter, D Price, D Riciutto, W Riley, A Sahoo, M Sprintsin, J Sun, H Tian, C Tonitto, H Verbeeck, S Verma. 2010. A model-data intercomparison of CO2 exchange across North America: Results from the North American Carbon Program site synthesis. Journal of Geophysical Research Biogeosciences VOL. 115, G00H0 doi:10.1029/2009JG001229*

**The interpretation of Figures 7 and 10 that the addition of additional data streams has little impact seems surprising. The authors suggest that the "Analysis LAI contracted considerably around the observations". In fact, the LAI analysis expands to accurately represent the prescribed observation error, as it should in a well-functioning system, but the ensemble mean approximately halves. A general interpretation of this would be that the model partitioning between leaf and other ABG carbon is incorrect. It is really impossible to draw any conclusion from Figure 10 without further diagnostics describing ensemble mean values, and error and bias across the 500 sites. The additional of more detailed diagnostics would also help to describe and assess how the system performs in modeling fluxes across CONUS. Currently, there is no discussion of where and when it does well or poorly. The authors indicate that 46 locations are selected to allow validation of fluxes (line 204), but no results of this are presented, and this is absolutely required.**

First, we are concerned the reviewer may be misinterpreting our Results as we in no way meant to imply that the addition of LAI had little impact on the Analysis carbon cycle estimates, as Figure 7 shows it clearly has a substantial impact. What Fig 10 shows is that despite this impact on the Analysis, the next year's forecast has similar uncertainty in LAI, which we interpret as suggesting that LAI forecast uncertainty is not dominated by initial condition uncertainty, but some combination of parameter, driver, and process error.

Second, as noted above, we want to reiterate that we specifically submitted this paper under "Development and technical paper", rather than as a research paper, to demonstrate a technical proof of concept. The goal of this paper is not to diagnose/validate the SIPNET model itself, as there are plenty of previous papers assessing the performance of both this model and its predecessor, PNET. We agree that this technical demonstration has uncovered a tendency for SIPNET to overestimate LAI when unconstrained, but this just reinforces our hesitation to overinterpret the overall C budget estimates.

Third, we also understand the reviewer's interest in seeing additional validation diagnostics, which are being developed in the companion paper being written by Morrison et al. Inclusion here would result in a quite large and sprawling paper, hence we deliberately decided to separate the technical note on system implementation from what is a full second paper's worth of material on validation. Furthermore, submission of this second paper has been delayed by our efforts to track down some of the model biases the reviewer notes. None of these biases have been related to the data assimilation workflow being presented in this paper (e.g. the LAI overestimation was related to VPD responses, not allocation). Rerunning our assimilation is costly (in terms of both compute and person-hours, the latter of which is limited as the grant supporting this analysis has concluded), so we would ask the reviewer's understanding in not requesting additional assimilation runs.

**The "Future Directions" section could be a lot more focused. It presents a long list of general possibilities that could be applicable to any terrestrial carbon cycle DA exercise – better models, more observations, higher resolution etc. Which, if any, are the authors actively pursuing with this system, and what is specific about their system that makes those developments particularly relevant?**
Thank you for this useful suggestion to sharpen our Future Directions section.
Current development is focused on an expansion of spatial scale (from CONUS to North America), an increase in spatial and temporal resolution, and the inclusion of additional data constraints. Next steps after that will be focused on assimilating disturbance.

At this point the future direction is focused on next technological steps for improving the performance of our data assimilation system (Line 476 - 488), additional data constraints and the justification for different data types (Line 496 - 525) and localization (526 -534).

**The appendix describing localization is interesting, and represents an additional development in methodology beyond that already described by Fer et al. and Raiho et al. Typically, it might be only expected to play an important role in model locations without observations – which by design is not the case here (assuming not many observations fail QC, which is needs to described and discussed as part of wider additional diagnostics as suggested above) but it's actually role here should be described, along with the implications of doing the localization by straightforward distance, when in actuality the model domain is being discretized along multiple environmental axes by the authors sampling technique. What is the implication of doing that for your localization approach?**

We agree with the reviewer that shifting the localization to environmental distance, rather than physical distance, would be an interesting area to explore. Doing so is beyond the current scope, but a useful Future Direction. It's worth noting that such a shift would be nontrivial because it would require developing a means for weighting the different environmental axes, each of which have different units, into a single "environmental distance".

We did not perform a run without spatial covariance, but agree with the idea implied by the reviewers that this would be useful Future Direction to assess the informational contributions of borrowing strength and where they occur in space and time. We have performed some initial analyzes assessing the spatial covariance structure over space and time, which does show interesting patterns of spatiotemporal synchrony, but the detailed analysis that would be required to say anything substantial was beyond the current scope (technical description) and thus was not included in the current manuscript.

References:
Albergel, C., Calvet, J.-C., Mahfouf, J.-F., Rüdiger, C., Barbu, A.L., Lafont, S., Roujean, J.-L., Walker, J.P., Crapeau, M., Wigneron, J.-P., 2010. Monitoring of water and carbon fluxes using a land data assimilation system: a case study for southwestern France. Hydrol. Earth Syst. Sci. 14, 1109–1124.

Albergel, C., Munier, S., Leroux, D.J., Dewaele, H., Fairbairn, D., Barbu, A.L., Gelati, E., Dorigo, W., Faroux, S., Meurey, C., Moigne, P.L., Decharme, B., Mahfouf, J.-F., Calvet, J.-C., 2017. Sequential assimilation of satellite-derived vegetation and soil moisture products using SURFEX_v8.0: LDAS-Monde assessment over the Euro-Mediterranean area. Geoscientific Model Development 10, 3889–3912.

Bacour, C., Peylin, P., MacBean, N., Rayner, P.J., Delage, F., Chevallier, F., Weiss, M., Demarty, J., Santaren, D., Baret, F., Berveiller, D., Dufrêne, E., Prunet, P., 2015. Joint assimilation of eddy covariance flux measurements and FAPAR products over temperate forests within a process-oriented biosphere model: JOINT ASSIMILATION OF NEE, LE, AND FAPAR. J. Geophys. Res. Biogeosci. 120, 1839–1857.

Boussetta, S., Balsamo, G., Dutra, E., Beljaars, A., Albergel, C., 2015. Assimilation of surface albedo and vegetation states from satellite observations and their impact on numerical weather prediction. Remote Sens. Environ. 163, 111–126.

Demarty, J., Chevallier, F., Friend, A.D., Viovy, N., Piao, S., Ciais, P., 2007. Assimilation of global MODIS leaf area index retrievals within a terrestrial biosphere model. Geophys. Res. Lett. 34, L15402.

Kumar, S.V., M. Mocko, D., Wang, S., Peters-Lidard, C.D., Borak, J., 2019. Assimilation of Remotely Sensed Leaf Area Index into the Noah-MP Land Surface Model: Impacts on Water and Carbon Fluxes and States over the Continental United States. J. Hydrometeorol. 20, 1359–1377.

Dorigo, W., Faroux, S., Meurey, C., Moigne, P.L., Decharme, B., Mahfouf, J.-F., Calvet, J.-C., 2017. Sequential assimilation of satellite-derived vegetation and soil moisture products using SURFEX_v8.0: LDAS-Monde assessment over the Euro-Mediterranean area. Geoscientific Model Development 10, 3889–3912.

Bacour, C., Peylin, P., MacBean, N., Rayner, P.J., Delage, F., Chevallier, F., Weiss, M.,Demarty, J., Santaren, D., Baret, F., Berveiller, D., Dufrêne, E., Prunet, P., 2015. Jointassimilation of eddy covariance flux measurements and FAPAR products over temperate forests within a process-oriented biosphere model: JOINT ASSIMILATION OF NEE, LE, AND FAPAR. J. Geophys. Res. Biogeosci. 120, 1839–1857.

Boussetta, S., Balsamo, G., Dutra, E., Beljaars, A., Albergel, C., 2015. Assimilation of surface albedo and vegetation states from satellite observations and their impact on  numerical weather prediction. Remote Sens. Environ. 163, 111–126.

Demarty, J., Chevallier, F., Friend, A.D., Viovy, N., Piao, S., Ciais, P., 2007. Assimilation of global MODIS leaf area index retrievals within a terrestrial biosphere model. Geophys. Res. Lett. 34, L15402.

Kumar, S.V., M. Mocko, D., Wang, S., Peters-Lidard, C.D., Borak, J., 2019. Assimilation of Remotely Sensed Leaf Area Index into the Noah-MP Land Surface Model: Impacts on Water and Carbon Fluxes and States over the Continental United States. J. Hydrometeorol. 20, 1359–1377

---

## Author Response (AR2)

Thank you for the comments. Here is our response to the following comments:

**Perhaps this sentence is a bit misleading, "All the code developed for this study can be found at https://doi.org/10.5281/zenodo.5557914" ? While the archive contains all the code specifically developed for the paper, I suspect that the point is that it also contains a lot of other code developed by others which is used, but not developed, in this work? Could the code that was actually developed for this manuscript be highlighted?**

In response to this comment, we modified the text in the "Code and Data availability" to reflect the true content of the Zendo repository. In addition, we added a new paragraph to this section, to highlight the exact codes developed through this project.

**I would usually expect all those whose code is first published in this manuscript to be included as authors on the manuscript, and their contribution described in the author contribution paragraph. In addition, I think it is appropriate to at least acknowledge the previous authors of code, by name, and outline their contribution in the manuscript, perhaps in the acknowledgments section - your ability to make progress rests directly on their earlier work.**

In response to this comment, we added two new co-authors to the list of authors and described their contribution to the "Author contribution" section. In addition, a new paragraph was added to the "Acknowledgements" section that acknowledges the contribution of whole PEcAn developer community and a small group of top developers in this community.